# Contracting with a Learning Agent

**Guru Guruganesh**[*]   **Yoav Kolumbus**[†]   **Jon Schneider**[*]   **Inbal Talgam-Cohen**[‡]

**Emmanouil-Vasileios Vlatakis-Gkaragkounis**[§]   **Joshua R. Wang**[*]   **S. Matthew Weinberg**[¶]

## Abstract

Real-life contractual relations typically involve repeated interactions between the principal and agent, where, despite theoretical appeal, players rarely use complex dynamic strategies and instead manage uncertainty through learning algorithms.

In this paper, we initiate the study of repeated contracts with learning agents, focusing on those achieving no-regret outcomes. For the canonical setting where the agent's actions result in success or failure, we present a simple, optimal solution for the principal: Initially provide a linear contract with scalar $\alpha > 0$, then switch to a zero-scalar contract. This shift causes the agent to "free-fall" through their action space, yielding non-zero rewards for the principal at zero cost. Interestingly, despite the apparent exploitation, there are instances where our dynamic contract can make *both* players better off compared to the best static contract.

We then broaden the scope of our results to general linearly-scaled contracts, and, finally, to the best of our knowledge, we provide the first analysis of optimization against learning agents with uncertainty about the time horizon.

## 1 Introduction

In the classic contract setting, a principal (she) incentivizes an agent (he) to invest effort in a project. The project's success depends stochastically on the effort invested. The incentive scheme, a.k.a. *contract*, is performance-based — it determines the agent's payment based on the project's outcome, rather than directly on the agent's effort. This gap between the agent's costly effort and the stochastic outcome creates *moral hazard*, and makes contract design a challenging problem.

Due to their immense importance in practice, the design of contracts has long been studied in Economics, forming a rich body of literature that was recognized by a Nobel prize in 2016. Recently, there has been a surge of interest in computational aspects of contract design, leading to the ongoing development of a new algorithmic theory of contracts (see, e.g., [1, 7, 8, 20, 21, 28, 29, 33, 30, 42, 43, 34, 51, 57, 62, 67]). Much of the computational research has focused on the classic, *one-shot* contract setting – the principal and agent share a single interaction, in which the principal offers a contract and the agent chooses a best-response action.[1] Yet, this overlooks the fact that in reality, *"most principal-agent relationships are repeated or long-term,"* i.e., the same agent exerts effort for the principal repeatedly over time [14]. The goal of this paper is to extend algorithmic contract

---

[*]Google Research, gurug@google.com, jschnei@google.com, joshuawang@google.com

[†]Cornell University, yoav.kolumbus@cornell.edu

[‡]Tel Aviv University, inbaltalgam@gmail.com

[§]Berkeley University, emvlatakis@berkeley.edu

[¶]Princeton University, smweinberg@princeton.edu

[1]Another setting that has been considered is a series of one-shot interactions with multiple agents, enabling the principal to learn the best contract for the agent population (see, e.g., [32, 23, 47, 69]). An exception is the work of [56], which studies a novel long-term principal-agent model tailored to afforestation. Their principal pays whenever a tree's state – a Markov chain – progresses; their agent responds based on state (not on learning).

38th Conference on Neural Information Processing Systems (NeurIPS 2024).

theory beyond the basic single-shot setting and into the realm of repeated contracts, applying a fresh perspective to contract design for repeated interactions.

Repeated contracts have been studied extensively in Economics. The literature explores many possible variations of how outcomes, actions, and contracts may evolve over time as the principal and agent interact (see, e.g., [64] and references therein; for surveys, see [24], [14, Chapter 10], and [55, Chapter 8]). The main theme of this literature is that the incentive problem grows significantly in complexity with repetition. First, the agent's action set becomes extremely rich, and optimizing over it is highly non-trivial. Second, the optimal contract itself typically becomes excessively complex – "*too complex to be descriptive or prescriptive for incentive contracting in reality*" [14]. Given this complexity, agents typically respond to repeated strategic interactions in practice in a manner consistent with no-regret learning [60, 61] (see Appendix A for a more detailed overview).

Building on this, we revisit the classic question of optimal contract design in a repeated setting, this time considering a no-regret learning agent. Hence, the main question we address in this work, referring to the *optimal dynamic contract*, is:

*If the principal knows that the agent is a no-regret learner, what contract sequence should she offer?*

## 1.1 Our Model and Contribution

**Optimizing against a no-regret learner.** We study the optimal dynamic contract in the following setting: A principal and agent interact over $T$ time steps for some large $T$. In each step $t \in [T]$, the agent takes a costly action as recommended by the no-regret learning algorithm, and the principal pays the agent according to the current contract and the action's outcome. The contracts can be modified by the principal over time dynamically (and adaptively). A simple benchmark is achieved by *not* modifying them, that is, simply repeating the optimal one-shot contract in each round. We refer to this as the *optimal static contract*. It is not hard to see that the principal's revenue in this case against a no-regret agent will essentially be the optimal static revenue (Observation I.1 in Appendix I).

Our main focus is on "mean-based" learning agents, who apply simple, natural and common learning algorithms, such as multiplicative weights [4], follow the perturbed leader [45, 50], or EXP3 [6]. Intuitively, mean-based algorithms consider the cumulative payoffs from each of the actions, and play actions which performed sub-optimally in the past with a low probability (see Section 2 for a precise definition, taken from [15]).

We also briefly consider more sophisticated agents who utilize *no-swap-regret*, rather than mean-based, learning (e.g., [13]). Against such agents, a crisp optimal strategy is immediate from previous work on general repeated games against learners [27, 59]: It is known that the best static solution is also the best dynamic one. In our context this means that no dynamic contract can achieve better than the optimal static contract (Observation I.2 in Appendix I). Since no-swap-regret learning also counts as a particular type of no-regret learning, this result also explains why focusing on mean-based learning is necessary for a separation result. Interestingly, we show that both players can be better off if the agent applies more naïve, mean-based learning (although, as expected, there are also many cases where the agent winds up worse off due to this interaction). Due to space constraints, we defer a more extended discussion of related work to Appendix A and highlight here our contributions.

**Our contribution.** In this paper, we give a clean, tractable answer to our main question as follows. When the agent's choice among $n$ actions can lead to success/failure of the project, the principal's optimal dynamic contract is surprisingly simple (especially compared to the optimal dynamic *auction* [15]): offer the agent one carefully-designed contract for a certain fraction of the $T$ rounds (both contract and fraction are poly-time computable), then switch to the zero contract (that is, pay the agent nothing) for the remaining rounds. In this setting, simple *linear* (single-parameter) contracts are optimal, and this is actually the only property required for our result. Our analysis thus generalizes to any setting where the principal utilizes only linear contracts. Note that much of the previous literature on algorithmic contract design has focused on the canonical contract setting with success/failure outcomes and/or on the ubiquitous class of linear contracts (see, e.g., [1, 7, 28, 29, 33, 56]).

Unlike the optimal dynamic *auction*, the optimal dynamic contract divides the welfare among the principal and agent. In fact, it can increase the utility of both players (and thus also the total welfare) in comparison to the optimal static contract. One interpretation of this result is the following. While a no-swap-regret algorithm is better for the agent against an adversarial player, an agent who commits to

using a mean-based algorithm allows the principal more freedom to dynamically implement outcomes of *common interest*. A similar advantage of simple no-regret learning over no-swap-regret was noted in a different context by [16].

Our main result also generalizes to settings with a rich set of outcomes beyond success/failure, as long as the principal changes the contract dynamically by scaling it ("single-dimensional scaling"). However, we also show that absent this single-dimensional scaling restriction, there exist principal-agent instances where the optimal dynamic contract does not take this form.
Equivalently, with non-linear contracts it is possible for the principal to do strictly better than offering the same contract for several rounds and then switching to the zero contract.

As our second main result, we identify and address the following gap in the current literature on optimizing against no-regret learners: Implicit in all our positive results, as well as in all known results in this literature (see Appendix A), is the assumption that the optimizer knows the time horizon $T$. We show that when there is (even limited) uncertainty about $T$, the principal's ability to use dynamic contracts to guarantee more revenue than the optimal static contract diminishes. We achieve this by characterizing the optimal dynamic contract under uncertainty of $T$, and showing that the principal's added value from being dynamic sharply degrades with an appropriate measure of uncertainty.

## 1.2 Illustrative Example

| Actions | "Failure" | "Success" |
|---|---|---|
| $a_1$: ($c_1 = 0/6$) | 1 | 0 |
| $a_2$: ($c_2 = 1/6$) | 1/2 | 1/2 |
| $a_3$: ($c_3 = 3/6$) | 0 | 1 |

Figure 1: A canonical contract setting in which a simple dynamic contract extracts higher expected revenue than the best static contract. The table entries show the outcome probabilities given the actions.

To demonstrate our model and findings, we give an example where a simple dynamic contract yields higher revenue than the best static contract. The analysis requires some familiarity with the basics of contract settings, which appear in the first paragraphs of Section 2 for completeness. Consider the setting in Figure 1. There are three actions with costs $c_1, c_2, c_3$ for the agent, leading, with the probabilities shown in the figure, to two outcomes—"failure" and "success"—with rewards 0 and 1 for the principal. Since there are two outcomes, w.l.o.g. we can consider linear contracts, which pay the agent $\alpha$ for success (leaving the principal with a payoff of $1 - \alpha$).

Under an optimal static linear contract, the agent must be indifferent either between actions 1 and 2 or between actions 2 and 3 (otherwise, the principal is overpaying for incentivizing an action). The indifference contracts are denoted by $\alpha_{1,2} = 1/3$ and $\alpha_{2,3} = 2/3$, respectively. These lead to the same expected utility for the principal, where the expectation is over the probability of success: $(1 - \alpha_{1,2}) \cdot \frac{1}{2} = (1 - \alpha_{2,3}) \cdot 1 = \frac{1}{3}$. That is, both $\alpha_{1,2}$ and $\alpha_{2,3}$ are optimal static contracts.

Now consider a principal interacting with a mean-based learning agent. The principal initially offers the contract $\alpha = \frac{2}{3} + \epsilon$ for $T/2$ time steps, with some small $\epsilon > 0$. The agent follows his mean-based strategy and plays action 3 (in a $1 - o(T)$ fraction of the time with high probability), which yields a utility of roughly $\alpha \cdot 1 - c_3 = \frac{1}{6}$ per step for the agent and $\frac{1}{3}$ per step for the principal. Subsequently, the principal switches to the zero contract for the remaining time steps. From the perspective of the agent, at the time of the switch the cumulative utilities of actions 2 and 3 are roughly $\frac{T}{12}$ (compared to zero from action 1). But in every step of the subsequent stage, the cumulative utility of action 3 is degraded by an amount of $\frac{1}{2}$ and the cumulative utility of action 2 is degraded by an amount of $\frac{1}{6}$. Thus the agent "falls" to action 2 and plays it until the last period $T$. The overall utility for the agent is approximately zero, and for the principal $\approx \frac{T}{2} \cdot \frac{1}{3} + \frac{T}{2} \cdot \frac{1}{2} = \frac{5}{12}T > \frac{1}{3}T$. The principal thus improves her utility by a factor of $\frac{5}{4}$ compared to the optimal static contract.

Figure 2 shows a graphical representation of the above dynamic contract. It turns out that this simple "free-fall" contract (see Definition 2.1) is also an optimal dynamic strategy for the principal, and in fact, this is not a special property of the current example. In Section 3 we show that for any linear contract game, there exists an optimal strategy with this form: offer a fixed contract for a period of $\lambda T$ steps and then switch to the zero contract. Moreover, we show that, surprisingly, this simple structure remains optimal also for general non-linear contracts, as long as their dynamics are characterized by a single scalar parameter. For details, see Appendix D.

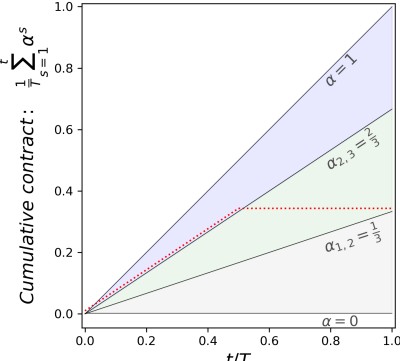 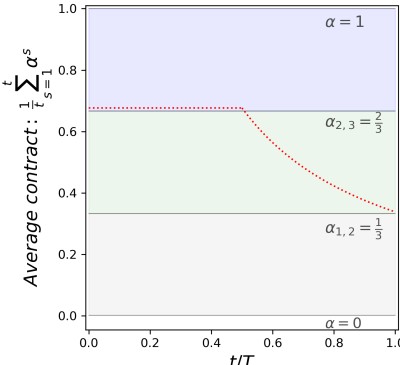

Figure 2: Two representations of the same dynamic contract, as applied to the contract setting described in Figure 1 and repeated for $T$ steps. The dotted red curve on the left describes the *cumulative* contract at time $t$ as a function of $t$, with both axes normalized by $T$. The shaded areas represent the mean-based best-response regions for the agent. The lines $\alpha_{1,2}$ and $\alpha_{2,3}$ are the *indifference curves* between these regions. The right diagram depicts the same dynamic contract as a function of the fraction of total time $t/T$, where here the vertical axis represents the *average* contract at time $t$. Pictorially, after steadily building the agent's incentives until time $T/2$, the principal cuts payments. During the remaining time, the agent "free-falls" through action regions.

## 1.3 Summary of Results and Roadmap

We initiate the study of repeated principal-agent problems with a learning agent; the key takeaways from our work are as follows:

**Theorem** (See Theorem 3.1 in Section 3.1). *In success/failure settings, as well as in arbitrary contract settings where the principal restricts to linear contracts, the optimal dynamic contract against a mean-based agent is a free-fall contract. This optimal dynamic contract can be efficiently computed.*

**Theorem** (See Theorem 3.2 in Section 3.2). *Consider the space of repeated principal-agent problems; in a subset of this space of positive measure, both the principal and agent achieve unboundedly better expected utilities from the principal's optimal dynamic contract compared to the optimal static contract.*

One takeaway from the previous theorem is that the principal and agent can *both* benefit when the agent commits to mean-based rather than no-swap-regret learning, in stark contrast to auctions (where a buyer committing to a mean-based strategy is left with zero payoff, because the auctioneer can extract the full surplus). We generalize our findings on the optimality of free-fall contracts to any dynamic contract with *single-dimensional scaling*.

**Theorem** (See Theorem D.1 in Appendix D). *In arbitrary contract settings, there is a free-fall contract that is optimal among dynamic contracts with single-dimensional scaling. A dynamic contract has single-dimensional scaling if it starts from an arbitrary contract $\mathbf{p} \in \mathbb{R}^m_{\geq 0}$, and at every time step $t \in [T]$ plays $\alpha^t \mathbf{p}$ for some scalar $\alpha^t \geq 0$.*

In Section 3.3 and Appendix G, we demonstrate that in absence of single-dimensional scaling, there may not exist a free-fall contract that is optimal among dynamic contracts.

Finally, we investigate the impact of an unknown time horizon when optimizing against a no-regret learner. To the best of our knowledge, we are the first that explore this aspect. Of course, there are standard techniques to help a learning *agent* can guarantee no-regret without knowing the time horizon (see e.g., [22, Section 2.3]). Typically, this agent manages to achieve with the same asymptotic (albeit with a larger constant factor) additive regret term when moving from the known to unknown time horizon setting. We show that the situation is different on the other side; uncertainty on the part of the *principal* regarding the time horizon significantly degrades her ability to outperform the best static contract. Whereas the agent's uncertainty was affecting their $o(T)$ factors, the principal's uncertainty degrades their $\Theta(T)$ factor; although some slight advantage is still always possible.

**Theorem** (See Theorems 4.2-4.3 in Section 4). *For any contract problem and error-tolerance parameter $\varepsilon > 0$, there exists is some minimum time uncertainty $\gamma$ so that for any minimum time horizon $\underline{T}$, no randomized dynamic contract can guarantee the principal a $(1 + \varepsilon)$ multiplicative advantage over the optimal static contract simultaneously for every time horizon $T$ in the range $[\underline{T}, \gamma\underline{T}]$. Conversely, for any contract problem and time uncertainty $\gamma > 1$, there is some nonzero error-tolerance parameter $\varepsilon > 0$ such that for a sufficiently large time horizon minimum $\underline{T}$, there is a randomized dynamic contract that can guarantee the principal a $(1 + \varepsilon)$ multiplicative advantage over the optimal static contract simultaneously for every time horizon $T$ in the range $[\underline{T}, \gamma\underline{T}]$.*

## 2 Model

We first present basic (non-repeated) contracts, and the class of linear contracts; a familiar reader may wish to skip to Sections 2.1-2.2 on repeated contract settings (discrete and continuous).

**Single-shot contract setting.** There are two players, a *principal* and an *agent*. The agent has a finite set $[n]$ of $n > 1$ *actions*, among which it chooses an action $a$ and incurs a corresponding *cost* $c_a \geq 0$ (in addition to $a$ we will use $i$ to index actions). W.l.o.g. the actions are sorted by cost $(c_1 < c_2 < ... < c_n)$ and the cost of the first (*null*) action is zero ($c_1 = 0$). There is a finite set $[m]$ of $m > 1$ possible *outcomes*, and every action $a$ is associated with a probability distribution $F_a \in \Delta^m$ over the outcomes. The null action leads with probability 1 to the first (*null*) outcome. Every outcome $o$ is associated with a finite *reward* $r_o \geq 0$ for the principal (in addition to $o$ we will use $j$ to index rewards). We assume w.l.o.g. that $r_1 \leq r_2 \leq ... \leq r_m$ and $r_1 = 0$. We denote the expected reward by $R_a = \mathbb{E}_{o \sim F_a}[r_o]$ for action $a$. As is standard we assume no dominated actions: $i$) if $c_a < c_{a'}$ then $R_a < R_{a'}$ and $ii$) for every action there exists a contract that uniquely incentivizes it. The contract setting (a.k.a. principal-agent problem) $(c, F, r) = (\{c_a, F_a\}_{a=1}^n, \{r_o\}_{o=1}^m)$ is known to both players.

**The game.** The game in the basic (non-repeated) setting has the following steps:
  (1) The principal commits to a *contract* $\mathbf{p} = (p_j)_{j=1}^m$, $p_j \geq 0$, where $p_j \leq p_{\max}$ is the non-negative amount the principal will pay the agent if outcome $j$ is realized.[2] In particular, $\mathbf{p}$ can be the *zero contract* in which $p_j = 0$ for all $j$.
  (2) The agent selects an action $a \in [n]$, unobservable to the principal, and incurs a cost $c_a$.
  (3) An observable outcome $o$ is realized according to distribution $F_a$. The principal receives reward $r_o$ and pays the agent $p_o$.
The principal thus derives a *utility* (*payoff*) of $r_o - p_o$, and the agent of $p_o - c_a$.

**Expected utilities and optimality.** In expectation over the outcomes, the utilities from contract $\mathbf{p}$ and action $a$ are $u_P(\mathbf{p}, a) = R_a - \mathbb{E}_{o \sim F_a}[p_o]$ for the principal, and $u_A(\mathbf{p}, a) = \mathbb{E}_{o \sim F_a}[p_o] - c_a$ for the agent. Summing these up we get the expected welfare $R_a - c_a$ from the agent's chosen action $a$. For a given contract $\mathbf{p}$, let $\mathsf{BR}(\mathbf{p}) = \arg\max_a u_A(\mathbf{p}, a)$ be the set of actions incentivized by this contract, i.e., maximizing the agent's expected utility (usually this will be a single element, but in the case of ties we include all actions in $\mathsf{BR}(p)$).[3] The goal of the contract designer is to maximize the principal's expected utility, also known as *revenue*. Such a contract is referred to as *optimal*.

**Linear contracts.** In a linear contract with parameter $\alpha \in [0, 1]$, the principal commits to paying the agent a fixed fraction (commission) $\alpha$ of any obtained reward. Thus by choosing action $a$, the agent gets expected utility $\alpha R_a - c_a$, and the principal gets $(1 - \alpha)R_a$. As $\alpha$ is raised from 0 to 1, the agent's expected utility is affected less by the action cost, and the agent's incentives align more with the principal's and with social welfare. This intuition is formalized by [33], showing that as $\alpha$ increases, the agent responds with actions that have increasing costs, increasing expected rewards, and increasing expected welfares.[4] The *critical* $\alpha$ at which the agent switches from action $i - 1$ to action $i$ (for $i > 1$) is denoted by $\alpha_{i-1,i} = (c_i - c_{i-1})/(R_i - R_{i-1})$, and is also referred to as an *indifference point* or *breakpoint*. For $i = 1$ we define $\alpha_{0,1} = 0$. Using this notation, for every linear contract $\alpha \in (\alpha_{i-1,i}, \alpha_{i,i+1})$, the agent plays action $i$. In the *linear contract setting*, the focus is on linear contracts and only such contracts are allowed.

---

[2]The non-negativity of the contractual payments is known as *limited liability* [49]. Without it – or some other form of risk aversion – the principal could simply "sell the project" to the agent, trivializing the problem.

[3]The standard tie-breaking assumption, according to which the agent breaks ties in favor of the principal, is less relevant here since we want to analyze all learning algorithms, regardless of how they break ties.

[4]We assume w.l.o.g. that for every action $a$ there is a linear contract $\alpha$ which uniquely incentivizes it (otherwise when focusing on linear contracts we may omit this action from the setting).

## 2.1 Repeated Contract Setting: Discrete Version

We study a repeated contract setting $(c, F, r, T)$, in which the above game $(c, F, r)$ is repeated for $T$ discrete rounds between the same principal and agent. The number of rounds $T$ is called the *time horizon*. The setting is known to both players,[5] who update the contracts and actions in each round. The outcomes of the actions are drawn *independently* per round (past outcomes affect future outcomes only through learning). Denote the contract, action, realized outcome and reward at time $t \in [T]$ by $\mathbf{p}^t, a^t, o^t, r^t$, respectively. The agent's payoff at time $t$ is $p_{o^t}^t - c_{a^t}$. The sequence $(\mathbf{p}^t)_{t=1}^T$ of contracts is called a *dynamic* contract, and the $T$ pairs $\{(\mathbf{p}^t, a^t)\}_{t=1}^T$ form the *trajectory of play*. We define the following class:

**Definition 2.1.** *A* free-fall *contract is a dynamic contract in which the principal offers a (single-shot) contract* $\mathbf{p}$ *for the first* $T' \leq T$ *rounds, and then offers the zero contract for the remaining rounds.*

**Learning agent.** The agent's approach to choosing an action is learning-based, by applying a no-regret algorithm (rather than based on myopic best-responding, as in the one-shot setting). Our analysis applies with *full feedback* on the performance of each action, where the agent observes the expected payoffs of all actions (whether taken or not — e.g. by observing someone else take that action), or with *bandit feedback*, where the agent observes only the achieved payoff of the action taken. A delicate issue is that, unlike the standard scenario of learning in games, the payments for each action are *stochastic*. Thus, the agent must not only learn which action to take, but also the expected payment from each action. When $T$ is large enough, the extra learning has a vanishing impact, and does not affect the analysis of players' utilities and strategies.

Our main focus is on the prominent family of *mean-based* algorithms. The idea behind mean-based algorithms is that they rarely pick an action whose current mean is significantly worse than the current best mean. There exist such algorithms with both full and bandit feedback that are mean-based and achieve no-regret. In our setting, let $u_i^t$ be the expected utility the agent would achieve from taking action $i$ at round $t$, and let $\sigma_i^t = \sum_{t'=1}^{t-1} u_i^{t'}$ represent the cumulative utility achievable from action $i$ up to time $t$ given the principal's trajectory of play. Then:

**Definition 2.2** ([15]). *A learning algorithm is* $\gamma(T)$-mean-based *if whenever* $\sigma_i^t < \sigma_{i'}^t - \gamma(T) \cdot T$, *then the probability that the algorithm takes action* $i$ *in round* $t$ *is at most* $\gamma(T)$. *We say an algorithm is* mean-based *if it is* $\gamma(T)$-mean-based *for some* $\gamma(T) = o(1)$.[6]

**Optimal dynamic contract.** The design goal in the repeated setting is to find an *optimal* dynamic contract: a sequence $(\mathbf{p}^t)_{t=1}^T$ that maximizes the total expected revenue against a learning agent (whether mean-based or no-swap-regret, where in either case we assume the worst-case such learning algorithm). In the linear contract setting, the sequence $(\alpha^t)_{t=1}^T$ is composed of linear contracts. If it maximizes the total expected revenue among all linear contract sequences, we say it is the optimal dynamic *linear* contract. We remark that it is without loss of generality to consider only linear contracts with[7] $\alpha \leq 1$.

Note that, as described here, the contract sequence is fixed by the principal at the beginning of the game. We refer to such a principal as *oblivious*. If the principal can choose $\mathbf{p}^t$ as a function of the agent's previous actions, we say the principal is *adaptive*. Our positive results (showing the principal can guarantee at least some amount of utility) hold even for oblivious principals, and our negative results hold even for adaptive principals.

**Optimal static contract.** In a repeated setting, a *static contract* is a sequence of contracts in which the same one-shot contract is played repeatedly. The repeated game with a static contract and a regret-minimizing agent is, in the limit $T \to \infty$, equivalent to the classic one-shot contract game with a best-responding agent (Observation I.1). A natural benchmark for dynamic contracts is thus the *optimal* static contract, in which the optimal one-shot contract is played repeatedly.

---

[5]In Section 4 we consider what happens when $T$ is unknown to the principal. The other parameters of the setting, if unknown, can be easily learned via sampling. No-regret learning is possible also with unknown $T$.

[6]Some small changes need to be made to this definition for the partial-feedback (bandits) setting – see Definition H.5 in Appendix H.3.

[7]This is a non-trivial consequence of our proof machinery. The proof appears as Observation I.3 in Appendix I for completeness.

## 2.2 Repeated Contract Setting: Continuous Version

To simplify the technical analysis, we now present a continuous version of our repeated contract setting. For the remainder of the paper we will primarily work in the continuous-time model. We emphasize that the reduction to continuous time is for simplicity, and that the key ideas of our proofs are unrelated to it and can be applied to the discrete version of our setting as well.

**Reduction to continuous time.** In [27], the authors consider the problem of strategizing against a mean-based learner in a repeated bi-matrix game, and show it reduces to designing dynamic strategies for a simplified continuous-time analogue (note that the choice of continuous-time analogue is tailored to mean-based learning – it is not intended to be a special case of a general discrete-continuous reduction against any learner). We pursue a similar reduction here, and show (in Theorem 2.4) how to reduce the problem of designing dynamic contracts in the discrete-time setting (Section 2.1), to a simpler problem in a continuous-time setting. We later extend the reduction to settings with an unknown time horizon (see Theorem H.3 in Section 4).

**Trajectories of continuous play.** In the continuous setting, rather than specifying the trajectory of play by a sequence of $T$ contracts and responses, we instead specify it by a finite sequence $\pi$ of tuples $\{(\mathbf{p}^k, \tau^k, a^k)\}_{k=1}^K$, each representing a "segment" of play where the principal plays a constant contract and the agent responds with a constant action. Here, each $\mathbf{p}^k \in \mathbb{R}_{\geq 0}^m$ represents an arbitrary contract, each $\tau^k \in \mathbb{R}_{\geq 0}$ represents the (fractional) amount of time that the principal presents this contract to the agent, and each $a^k \in [n]$ represents the action the agent takes during this time. In the linear contract setting, we use the notation $\alpha^k$ instead of $\mathbf{p}^k$. We sometimes refer to $\pi$ as a contract, by which we mean the dynamic contract composed of $\mathbf{p}^1, \ldots, \mathbf{p}^K$ for segments of length $\tau^1, \ldots, \tau^K$.

To form what we call a *valid* trajectory of play against a mean-based learner, the responses $a^k$ of the agent must satisfy certain constraints. Let

$$\mathcal{T}^k = \sum_{k'=1}^k \tau^{k'}; \quad \overline{\mathbf{p}}^k = \sum_{k'=1}^k (\mathbf{p}^{k'} \tau^{k'}) / \mathcal{T}^k$$

be the total duration of the first $k$ segments, and the average contract offered by the principal for the first $k$ segments, respectively. Then each $a^k$ (for $k > 1$) must satisfy $a^k \in \mathsf{BR}(\overline{\mathbf{p}}^{k-1})$ and $a^k \in \mathsf{BR}(\overline{\mathbf{p}}^k)$. In words, $a^k$ must be a best-response to the historical average contract at both the beginning and end of segment $k$ (and therefore also throughout segment $k$).

The following is a continuous analogue of Definition 2.1.

**Definition 2.3.** *A* free-fall *trajectory $\pi$ is a game trajectory in which $\mathbf{p}^k = \mathbf{0}$ for all $k > 1$.*

**Optimal trajectory.** The expected utility of the principal along trajectory $\pi$ is given by

$$\mathsf{Util}(\pi) = \frac{\sum_{k=1}^K \tau^k u_P(\mathbf{p}^k, a^k)}{\mathcal{T}^K}.$$

Let $U^\star = \sup_\pi \mathsf{Util}(\pi)$, where the sup runs over all valid trajectories of arbitrary finite length. We can think of $U^\star$ as the maximum possible expected utility of the principal in the continuous setting game. The following theorem (a direct analogue of Theorem 9 in [27]) connects $U^\star$ to what is achievable by the principal in our original discrete-time game.

**Theorem 2.4.** *Fix any repeated principal-agent problem with $T$ rounds, and let $U^\star$ denote the optimal expected utility of a principal in the continuous analogue. Then:*
1. *For any $\varepsilon > 0$, there exists an oblivious strategy for the principal that gets at least $(U^\star - \varepsilon)T - o(T)$ expected utility for the principal against an agent running any mean-based algorithm $\mathcal{A}$.*
2. *For any $\varepsilon > 0$, there exists a mean-based algorithm $\mathcal{A}$ such that no (even adaptive[8]) principal can get more than $(U^\star + \varepsilon)T + o(T)$ expected utility against an agent running $\mathcal{A}$.*

The proof of Theorem 2.4 closely follows the proof in [27] and is deferred to Appendix H.

---

[8]In the partial-feedback (bandit) setting, this result only holds for *deterministic* adaptive principals and not for randomized adaptive principals (with full-feedback, it holds for either); see Appendix H.3 for further discussion.

One important thing to note about Theorem 2.4 is that the first part is constructive. In fact, the discrete-time strategy for the principal corresponding to a trajectory $\pi$ is essentially the straightforward extrapolation, which plays each contract $\mathbf{p}^k$ for $\frac{\tau^k}{\mathcal{T}^K}T$ rounds (although a slight perturbation is necessary to account for segments with a non-unique best-response). This means that when we show, in Theorem 3.1, that the utility-optimizing $\pi$ for $U^\star$ takes the form of a free-fall trajectory, we are simultaneously showing that a free-fall dynamic contract is asymptotically optimal in the original discrete-time setting.

Note that all the above definitions (and the reduction of Theorem 2.4) extend to the specific case where the learner is only allowed to use *linear contracts*. In this setting, we will write $\overline{\alpha}^k = \sum_{k'=1}^{k} \alpha^{k'} \tau^{k'} / \sum_{i=1}^{k} \tau^{k'}$ in place of $\overline{\mathbf{p}}^k$.

# 3 Linear Contracts

In this section we focus on the case where the principal restricts to using only linear contracts in every step of the interaction with the agent (one example is when there are $m = 2$ outcomes, such as success and failure; in this case, arbitrary contracts can be described as linear contracts). We begin, in Section 3.1, by showing that without loss of generality, optimal dynamic contracts take the form of free-fall contracts, and in Appendix D we generalize this result to a broader class of general contracts with single-dimensional scaling. Then, in Section 3.2, we analyze the implications of optimal free-fall contracts on the welfare and on the agent's utility. In particular, we show that dynamic contracts that are optimal for the principal can improve the utilities for both players compared to their utilities under the best static contract. Finally, in Section 3.3, we show that for unrestricted dynamic contracts, free-fall contracts may no longer be optimal.

## 3.1 Free-Fall Contracts are Optimal Linear Contracts

The following theorem shows that free-fall contracts are optimal dynamic linear contracts.

**Theorem 3.1.** *Let $\pi$ be any linear dynamic contract. Then, there exists a free-fall linear contract $\pi'$ where $\mathsf{Util}(\pi') \geq \mathsf{Util}(\pi)$, and which can be computed in time polynomial in the problem size.*

The proof is deferred to Appendix B and hinges on applying a sequence of "rewriting" rules which allow us to gradually transform any given linear dynamic contract $\pi$ with a free-fall linear contract $\pi'$. At a high level, the crux of the proof is that any segment of the trajectory can be thought of as a combination of "stalling" at the current action and "falling" to the action below. Under linear contracts, the principal prefers to stall when a higher action is being induced. Grouping together all the stall at the highest action used exactly results in a free-fall contract.

## 3.2 Implication to Welfare and Agent's Utility

In the example shown in Section 1.2, the free-fall dynamic manipulation that the principal made degraded the overall welfare, and all the added profits for the principal were at the expense of the agent. We demonstrate that this is not always the case; there are other scenarios where dynamic manipulations where the principal, optimizes her revenue, can actually be Pareto improvements over the best static contract, increasing the overall welfare.

**Example** (Welfare improvement). *Consider the setting depicted in Figure 1 with a slight variation where the cost of action 2 is $1/2 + \epsilon$, with $\epsilon < 1/(2T)$. In this case, the best static contract incentivizes action 1 and yields a utility of $1/3$ for the principal and zero for the agent. However, the best dynamic contract remains the same as in the previous analysis: it starts by incentivizing action 2 for a period of $\frac{2}{3}T$ steps and then transitions to action 1 by offering zero payments for the remaining time. This results in a utility of $\frac{5}{12}T$ for the principal and zero for the agent, thereby increasing welfare by a factor of $5/4$ without altering the agent's utility.*

Next, we show the existence of "win-win" scenarios where optimal dynamic contracts can enhance the payoffs for both the principal and the agent compared to the best static contract. The improvement in welfare can be substantial, reaching as much as $\mathcal{O}(n)$, essentially achieving full welfare. Specifically, we establish that the multiplicative gap between the utilities of the best static contract and those of the best dynamic contract can be $\mathcal{O}(n)$ for the principal's utility and $\mathcal{O}(\log(n))$ for the agent's utility.

**Theorem 3.2** (Win-win optimal dynamic contracts). *There exist repeated contract settings where an optimal dynamic contract improves expected welfare by a $\Theta(n)$ multiplicative factor compared*

*to the best static contract, and where the agent's expected utility improves by a factor of $\Theta(\log(n))$. Moreover, these settings have a positive measure in the space of repeated contract games.*

The idea of the proof is to look at games where the values for the principal when incentivizing each action are similar, but the actions differ significantly in terms of welfare. Then, by investing a small amount of additional payment in the early stages of the game (compared to the best static contract), the principal can incentivize the agent to substantially improve welfare, initially in the form of higher profits for the agent. This added welfare is then shared between the players during the free-fall stage of the dynamic.

The proof is deferred to Appendix C. One interesting point about the example presented in the proof is that if the agent had used a "smarter" learning algorithm that guaranties low swap regret, then the outcome of the best static contract would have been obtained (see Observation I.2 in the appendix, following the analysis of [27]). The agent in this case would have had lower utility. That is, using a better algorithm leads to a worse outcome! The explanation for this counter-intuitive result is that a mean-based regret-minimizing algorithm is only guaranteed to approach the set of Coarse Correlated Equilibria (CCE),[9] whereas no-swap-regret dynamics must approach the set of Correlated Equilibria (CE, a subset of the set of CCE's). There are games in which some CCE distributions of play give higher utilities to the players than all CE distributions (see e.g., [36, 53] for examples in auctions, and [16] for a related example in general games). In other words – committing to use an algorithm with weaker worst-case guarantees yields better (non-worst-case) results.

### 3.3   General Contracts and Free-Fall

Unlike in the linear contract setting and the single-dimensional scaling setting, free-fall contracts are *not* optimal in the general contract setting. We provide an example outlining this in Appendix G. In fact it is an open question whether the the optimal dynamic contract is computable.

## 4   Unknown Time Horizon

Up until now, the principal has been able to take advantage of precisely knowing the time horizon. Notably, this assumption of knowledge of the time horizon underlies all prior theoretical results in the literature on optimization against learning algorithms. In this section, we explore what happens when the principal only approximately knows this parameter. We will consider the case where the principal knows that the time horizon $T$ falls into some range $[\underline{T}, \overline{T}]$, and wants to guard against a worst-case choice of time horizon from that range. What are the trade-offs between the time uncertainty and how much additional principal utility we can get over the best static contract? To explore these concepts precisely, we introduce the following definition:

**Definition 4.1.** *Suppose we have a principal-agent problem $(c, F, r)$. Let $R_\star$ be the single-round profit of the optimal static linear contract for this problem. We say that a pair $(\epsilon, \gamma)$ is feasible with respect to $(c, F, r)$ if for all sufficiently large time horizons $\underline{T}$, there exists a (potentially randomized) principal algorithm $A$ such that the (expected) profit of $A$ at any time $t \in [\underline{T}, \overline{T} = \lceil \underline{T}\gamma \rceil]$ is at least $(1 + \epsilon)tR_\star$ (and infeasible with respect to $(c, F, r)$ otherwise).*

The last part of our results is Theorem 4.2: for every principal-agent problem and any error-tolerance $\varepsilon > 0$, it is impossible to indefinitely maintain an $\varepsilon$ advantage over the optimal static contract. To be more precise, when $\gamma$ is $\exp(\Omega(1/\varepsilon))$ we know that the instance has become $(\varepsilon, \gamma)$ infeasible (for some instances, this infeasibility transition may occur earlier). We argue this via a potential function; in order to stay a constant factor ahead of the optimal static contract, the principal must be constantly giving up potential. To complement this result, in Theorem 4.3, we show that all time ratios $\gamma \geq 1$ permit some advantage $\varepsilon$ over the optimal static contract. We manage to achieve $\varepsilon$ at least $\Omega(1/(\text{poly } \gamma))$ for all problems where the optimal dynamic contract (with known time horizon) outperforms the optimal static contract. These ideas are captured in the theorems below:

**Theorem 4.2.** *Suppose we have a principal-agent problem $(c, F, r)$. For every $\epsilon > 0$, there exists a $\underline{\gamma}$ such that $(\epsilon, \gamma)$ is infeasible with respect to $(c, F, r)$ for all $\gamma \geq \underline{\gamma}$.*

---

[9]In fact, for mean-based algorithms there is a stricter characterization of the set of equilibria to which they may converge [53], but the above explanation still holds.

**Theorem 4.3.** *Suppose we have a principal-agent problem $(c, F, r)$. If there exists a $\underline{\gamma}$ such that for any $\epsilon > 0$ and $\gamma \geq \underline{\gamma}$, $(\epsilon, \gamma)$ is infeasible, then there are no dynamic strategies that outperform the optimal static linear contract.*

The proof of Theorem 4.2 is in Appendix E; Theorem 4.3, Appendix F. The key ideas are as follows. For Theorem 4.2, we construct a potential function which assigns a value to the current time-averaged linear contract, and show that any principal is forced to slowly sacrifice this potential as the possible time horizon gap grows. For Theorem 4.3, any dynamic strategy that outperforms the optimal static linear contract can also be made to either start or end at the optimal static linear contract. This allows us to pad such a strategy to last for a longer amount of time by adding a segment that just stalls at the optimal static linear contract. One technical issue we have to handle is that trajectories must be evaluated over the interval $[\frac{1}{\gamma}\mathcal{T}^K, \mathcal{T}^K]$ instead of at a single time; this multidimensionality means we must now consider distributions of trajectories (see Appendix H).

## 5   Conclusion

In this paper, we provide a clean and tractable answer to our main question. When the agent's choice among $n$ actions can lead to the success or failure of a project, the principal's optimal dynamic contract is surprisingly simple. Specifically, the principal should offer a carefully designed contract for a certain fraction of the $T$ rounds (both the contract and the fraction are poly-time computable), then switch to a zero contract (i.e., pay the agent nothing) for the remaining rounds. Our main result also generalizes to settings with a rich set of outcomes beyond success/failure, as long as the principal changes the contract dynamically by scaling it ("single-dimensional scaling"). However, we show that without this single-dimensional scaling restriction, there exist principal-agent instances where the optimal dynamic contract does not take this form. In these cases, with non-linear contracts, the principal can do strictly better than offering the same contract for several rounds before switching to a zero contract.

As our second main result, we address a significant gap in the current literature on optimizing against no-regret learners: the assumption that the optimizer knows the time horizon $T$. We show that when there is uncertainty about $T$, even if limited, the principal's ability to use dynamic contracts to guarantee more revenue than the optimal static contract diminishes. We characterize the optimal dynamic contract under uncertainty of $T$, demonstrating that the principal's added value from being dynamic sharply degrades with an appropriate measure of uncertainty.

**Open Problems.**   The computational study of repeated contracts, particularly with learning agents, raises many open questions. These include determining the optimal dynamic contract when the principal is not restricted to one-dimensional dynamics, and the computational complexity of finding it. Additionally, it involves identifying the optimal dynamic contract against a learning agent with a hidden type, thereby unifying our contract model with the auction model of [15]. Another intriguing area is understanding what the optimal dynamic contract would be against a team of multiple learning agents. Finally, it is crucial to explore the effects on welfare and utilities when there are two learning players, rather than a learner and an optimizer.

### Acknowledgments and Disclosure of Funding

This work received funding from the European Research Council (ERC) under the European Union's Horizon 2020 research and innovation program (grant No.: 101077862, project: ALGOCONTRACT, PI: Inbal Talgam-Cohen), by the Israel Science Foundation (grant No.: 3331/24), by the NSF-BSF (grant No.: 2021680), by the NSF (grant No.: CCF-1942497, PI: S. Matthew Weinberg) and by a Google Research Scholar Award. During Professor Weinberg's development of this paper, he participated as an expert witness on behalf of the State of Texas in ongoing litigation against Google (the "Google Litigation").

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

# Appendix

## A Further Related Work

**Introduction to the Problem.** Consider the following motivating examples: a worker joining a new team, a student starting an internship, or a junior professor joining a committee. These agents initially face uncertainty about the required effort and what constitutes good performance. They must decide when to exert more effort and when to reduce it. Peer assessments introduce additional uncertainty and noise. Furthermore, the environment is dynamic, with the value of certain outcomes changing over time. Each agent encounters an implicit and evolving system of incentives that they must adapt to through repeated interactions. This pattern is prevalent in many real-life contractual relationships and is increasingly relevant to AI agents handling complex, open-ended, and computationally intensive tasks. For details to the aforementioned examples, the interested reader can see [41] and references therein. In settings like credit scoring, the evaluation system creates incentives for the agent while remaining opaque to prevent gaming, forcing the agent to act under uncertainty [38].

**Simplifying Contracts.** Given this complexity, one line of work focuses on identifying settings where simple contracts suffice. Notably, [48] assume constant absolute risk aversion (CARA) utilities and Brownian motion of the output, examining a single payment at the end of the contractual relationship based on all outcomes. Another approach involves deliberately vague contracts, leaving agents uncertain about performance-based compensation (e.g., [3, 11, 31]). [44] explore how to learn an agent's private type through online principal-agent interaction and contract menus. [10] study principal-agent problems over MDPs, where a budgeted principal offers additional rewards, and the agent selects the MDP policy selfishly, without learning. Thus, a naturally arising question is:

> *How should an agent choose their actions in a contractual relationship*
> *involving uncertainty and recurrent interactions?*

Our algorithmic perspective introduces a novel, learning-based approach to address the complexity of repeated contracts, leveraging no-regret and general mean-based agents. Below, we discuss why learning methods are natural choices for agents' responses in the context of existing literature.

**Optimizing Against No-Regret Learners.** From an econometric perspective, agents often respond to repeated strategic interactions in auctions in ways consistent with no-regret learning [60, 61]. Inspired by these findings, [15] explore algorithmic mechanism design, demonstrating that no-regret learning methods are natural responses for agents. No-regret learning has been extensively studied in repeated games (e.g.,[2, 5, 12, 16, 35, 46, 54, 58, 68, 39, 40, 63, 66]), auctions and economic interactions (e.g.,[25, 19, 37, 53, 52]), and Stackelberg security games (e.g., [9]). For a comprehensive overview, see [65]. By assuming agents employ no-regret learning instead of complex strategic reasoning, we propose a new approach to repeated contracting.

**Optimizing Against Mean-Based Learners.** Finding an optimal dynamic strategy against a mean-based learner in general games remains an open problem. [27] show an equivalence between this problem and an $n$-dimensional control problem, where $n$ is the number of actions available to the agent. Non-trivial optimization against a mean-based learner has been achieved only in repeated auction settings, where [15] demonstrate that the designer can extract full welfare as revenue. [26, 18] extend this to prior-free auction settings and multiple agents. However, even for a single agent, the optimal auction strategy, involving alternating between second-price auctions and charging large payments, is impractical and not intended to guide practice [18]. [19] study mechanisms for no-regret agents, incorporating principal learning to avoid common prior assumptions in economic design problems.

## B Proof of Theorem 3.1 (Optimal Dynamic Linear Contract)

**Proof overview.** We will present a series of "rewriting" rules, which will allow us to replace a given dynamic contract $\pi$ with a simpler, more constrained, dynamic contract $\pi'$ with utility at least as large as $\pi$. At the conclusion of our sequence of rewriting steps, we will see that our contract takes the form of a free-fall contract, thus implying that there is an optimal free-fall contract.

We begin not with a rewriting rule, but instead a general observation about the structure of dynamic linear contracts — namely, that it is impossible for an agent to "skip over" an action. That is, if an agent is playing action $i$ at some point, and action $j$ at some later point, there must exist segments of

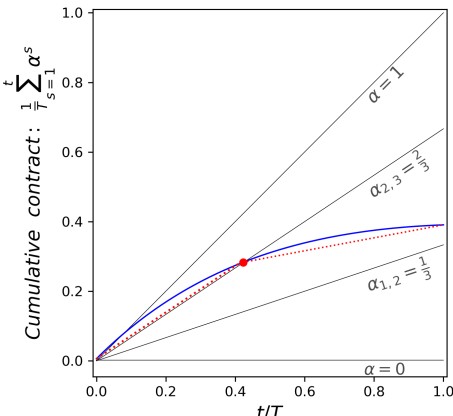

Figure 3: An illustration of Lemma B.2. The plot shows the cumulative contract over time for the contract game shown in Figure 1, repeated to $T$ steps, where both axes are normalized by $T$. The lemma shows how arbitrary dynamic strategies, as the one shown in the blue curve, can be re-written as piecewise stationary strategies, depicted in the dotted red curve, inducing similar behavior by the agent and the same utilities.

non-zero duration where the agent plays each of the intermediate actions between $i$ and $j$. Formally, we can write this as follows.

**Lemma B.1.** *If $\pi = \{(\alpha^k, \tau^k, a^k)\}$ is a dynamic linear contract, then for any $k$, $|a^k - a^{k+1}| \leq 1$ (i.e., in any two consecutive segments, the learner's action can change by at most one).*

*Proof.* Note that $a^k$ and $a^{k+1}$ must both be best responses to the average historical contract $\overline{\alpha}^k$, i.e., $a^k, a^{k+1} \in \mathsf{BR}(\overline{\alpha}^k)$. Since $\mathsf{BR}(\alpha)$ is always of the form $\{i\}$ or $\{i, i+1\}$, the conclusion follows. $\square$

Note that the proof of Lemma B.1 relies on the "linear topology" of the best-response regions in Figure 2 (i.e., any non-zero boundary between best-response regions connects two consecutive actions of the agent). This property is *not* true for general contracts or general games; however, we will later see that Lemma B.1 also holds for the class of **p**-scaled contracts introduced in Appendix D.

We now proceed to introduce our rewriting rules. The first rewriting rule we present is very general (and in fact applies to any game): we will show that without loss of generality, no two consecutive segments of a dynamic contract induce the same action for the learner. Intuitively, for any time interval in which a mean-based agent plays a single action, we can replace the contracts in this interval with their average and obtain overall a revenue-equivalent dynamic contract. Formally, we can phrase this as follows.

**Lemma B.2.** *Let $\pi$ be any linear dynamic contract. Then there exists a linear dynamic contract $\pi'$ such that $\mathsf{Util}(\pi') \geq \mathsf{Util}(\pi)$ and no two consecutive segments of $\pi'$ share the same agent action ($a^k \neq a^{k+1}$).*

*Proof.* Let $\pi = \{(\alpha^k, \tau^k, a^k)\}_{k=1}^{K}$ be any linear dynamic contract. Assume that for some $k$, $a^k = a^{k+1} = a$. Then the linear dynamic contract $\pi'$ formed by replacing the two segments $(\alpha^k, \tau^k, a)$ and $(\alpha^{k+1}, \tau^{k+1}, a)$ with the single segment $((\alpha^k \tau^k + \alpha^{k+1} \tau^{k+1})/(\tau^k + \tau^{k+1}), \tau^k + \tau^{k+1}, a)$ has the property that $\mathsf{Util}(\pi') \geq \mathsf{Util}(\pi)$. To see this, observe that the same action $a$ is played throughout the entire interval, and the average payout to the agent is the same. Therefore, in fact we have $\mathsf{Util}(\pi') = \mathsf{Util}(\pi)$. It only remains to confirm that this is still a valid dynamic contract (i.e., that each prescribed action is still a best response in the corresponding segment).

To see this, observe first that the cumulative contract at the start (respectively, end) of the merged segment in $\pi'$ is the same as the start of segment $k$ (respectively, end of segment $k+1$) in $\pi$. Therefore, all segments before and after the merged segment are still correct. To confirm the merged segment, we need only confirm that $a$ is a best response on the merged segment in $\pi'$ using the fact that it was a best response in both segments $k$ and $k+1$ in $\pi$.

For this, let $\alpha_0$ denote the cumulative contract after the first $k-1$ segments, $\alpha_1$ denote the cumulative contract after the first $k+1$ segments, $\alpha'(t)$ denote the cumulative contract of $\pi'$ during a time $t$ in the merged segment, and $\alpha(t)$ denote the cumulative contract of $\pi$ during a time $t$ in segments $k$ or $k+1$. Observe first that for every $x$ between $\alpha_0$ and $\alpha_1$, there is some $t$ such that $\alpha(t) = x$. Because $a$ is a best response on the entire segments $k$ and $k+1$, this means that $a$ is a best response to $x$ for all $x$ between $\alpha_0$ and $\alpha_1$. Moreover, observe that $\alpha'(t)$ lies between $\alpha_0$ and $\alpha_1$ for all $t$. Therefore, $a$ is indeed a best response to $\alpha'(t)$ for all $t$ in the merged segment, and the dynamic contract is valid.

By repeatedly applying this merging of segments, we can obtain a linear dynamic contract $\pi'$ satisfying the constraints of the lemma. $\qquad \square$

Figure 3 illustrates the above lemma graphically. The figure displays the cumulative contract over time for the contract game depicted in Figure 1. The blue curve represents the trajectory of an arbitrary dynamic contract strategy under which the agent's best response is to take action 3 until time $t/T \approx 0.425$, and then take action 2 in the remaining time. The crossing point between the best response regions is marked with a red dot. Lemma B.2 demonstrates that we can replace the blue trajectory with the simpler trajectory depicted in red. In this red trajectory, every region between two consecutive $\alpha$ values is crossed by a single linear segment (i.e., a piecewise-stationary trajectory), resulting in the same behavior by the agent and the same revenue.

Our second rewriting rule is specific to linear contracts. It shows that for every linear contract in which the agent is indifferent between two actions, it is beneficial for the principal to shift the contract infinitesimally so that the agent prefers the action with the higher expected reward.

**Lemma B.3.** *Let $\pi = \{(\alpha^k, \tau^k, a^k)\}_{k=1}^K$ be a dynamic linear contract where during segment $k$ the agent is indifferent between actions $i$ and $i+1$ (i.e., $\mathsf{BR}(\overline{\alpha}^{k-1}) \cap \mathsf{BR}(\overline{\alpha}^k) \supseteq \{i, i+1\}$), but $a^k = i$. If we form $\pi'$ by replacing $a^k$ with $i+1$, then $\mathsf{Util}(\pi') \geq \mathsf{Util}(\pi)$ (the principal always prefers that the agent plays the action with higher expected reward).*

*Proof.* Since actions in the linear contract setting are sorted by increasing value of expected reward, we have that $\mathsf{Util}(\pi') - \mathsf{Util}(\pi) = \frac{\tau^k}{\mathcal{T}^K}\left(u_P(p^k, i+1) - u_P(p^k, i)\right) = \frac{\tau^k}{\mathcal{T}^K}\left(R_{i+1} - R_i\right)(1 - \alpha^k) \geq 0$. $\qquad \square$

Note that the principal can implement the change in the agent's action in Lemma B.3 by simply increasing their payment to the agent by an arbitrarily small amount – this incentivizes the agent to break ties in favor of the action with larger expected reward (which is the action labeled with a larger number). The fact that the principal can implement this change also follows as a direct consequence of the discrete-to-continuous reduction of Theorem 2.4.

By applying the above two rewriting rules (Lemmas B.2 and B.3) along with our observation in Lemma B.1, we can establish our third rewriting rule: it is always possible to rewrite a dynamic contract so that the sequence of actions is a consecutively decreasing sequence.

**Lemma B.4.** *Let $\pi$ be any dynamic linear contract. Then there exists a dynamic linear contract $\pi' = \{(\alpha^k, \tau^k, a^k)\}_{k=1}^K$ with $\mathsf{Util}(\pi') \geq \mathsf{Util}(\pi)$ and where $a^1, a^2, \ldots, a^K$ is a decreasing sequence of consecutive actions (i.e., $a^k = a^1 - (k-1)$).*

*Proof.* Apply the two rewriting rules in Lemmas B.2 and B.3 to $\pi$ until it satisfies the post-conditions of both lemmas (so, no two consecutive segments incentivize the same action, and any segment on a best-response boundary incentivizes the higher-reward action). Since Lemma B.1 implies that consecutive segments cannot skip over an action, this means that every two consecutive actions under $\pi$ are consecutive: either the agent switches to the next higher action or the next lower action each time step. We therefore just must show that any dynamic contract where the agent increases their action can be rewritten as a decreasing contract with at least same payoff.

Consider the first segment in $\pi$ where the agent switches to a larger action, that is, the smallest $k$ such that $a^{k+1} = a^k + 1$. Let $a^k = j$ (so $a^{k+1} = j + 1$). Note that the agent must be indifferent between actions $j$ and $j+1$ at the end of the $j$th segment (i.e., $\{j, j+1\} \subseteq \mathsf{BR}(\overline{\alpha}^k)$).

There are two cases: either segments $k$ and $k+1$ are the first two segments of the dynamic contract $\pi$ (i.e., $k = 1$), or there exists a $(k-1)$st segment. In the first case, the agent is indifferent between

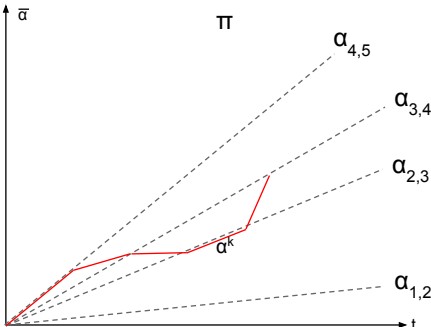

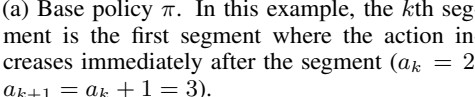

(a) Base policy $\pi$. In this example, the $k$th segment is the first segment where the action increases immediately after the segment ($a_k = 2$, $a_{k+1} = a_k + 1 = 3$).

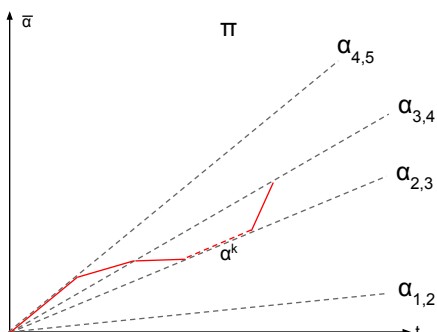

(b) Since $a_{k-1} = a_{k+1}$, the $k$th lies along the best response boundary $\alpha_{2,3}$, and its existence violates Lemma B.3 (we can rewrite it as a segment with $a_k = 3$, and then further collapse these segments via Lemma B.2).

Figure 4: Illustrations for the proof of Lemma B.4.

actions $j$ and $j + 1$ for the entire first segment (from time $0$ to $\tau^1$), but plays action $j$. This contradicts the fact that $\pi$ cannot be reduced further by Lemma B.3.

In the second case, the $(k-1)$st segment must incentivize action $j + 1$ for the learner (since the sequence of actions is decreasing up until segment $k$). But this means that the agent must be indifferent between actions $j$ and $j + 1$ also after the $(k-1)$st segment, and thus for the entirety of the $k$th segment ($\{j, j+1\} \subseteq \mathsf{BR}(\overline{\alpha}^{k-1})$). Since the agent plays $j$ during the $k$th segment, this also contradicts the fact that $\pi$ cannot be reduced further by Lemma B.3 (see Figure 4 for an example of this reduction). $\qquad\square$

We are now almost done – Lemma B.4 shows we can rewrite any dynamic contract so that the agent descends through their action space. We now need only show that the principal should abruptly switch to offering the zero contract after the first segment (instead of slowing the rate of descent through these regions by offering a positive contract). We do this in our final rewriting lemma.

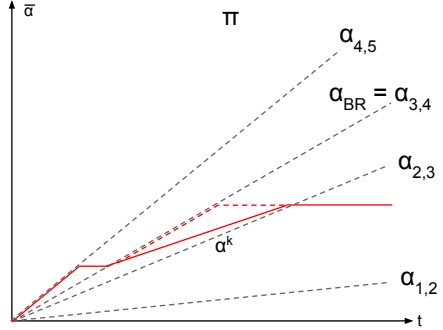

(a) Base policy $\pi$. The $k$th segment is the first non-free-fall segment; we decompose it into a segment $\alpha = \alpha_{BR} = \alpha_{3,4}$ and a free-fall segment with $\alpha = 0$.

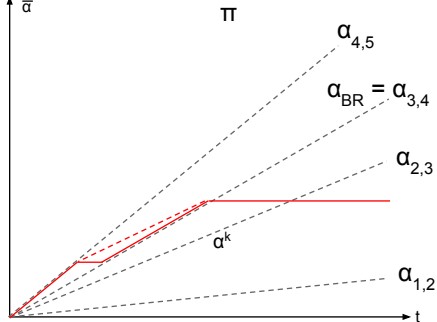

(b) The segment with $\alpha = \alpha_{BR}$ can be rewritten via Lemma B.3 to lie in region 4, where it can then be further combined with earlier segments via Lemma B.2. This moves the first non-free-fall action earlier in $\pi$.

Figure 5: Illustrations for the proof of Lemma B.5.

**Lemma B.5.** *Let $\pi$ be a dynamic linear contract where the agent plays a decreasing sequence of actions. Then there exists a free-fall linear contract $\pi'$ with $\mathsf{Util}(\pi') \geq \mathsf{Util}(\pi)$.*

*Proof.* Let $\pi = \{(\alpha^k, \tau^k, a^k)\}_{k=1}^K$ (with $a^k$ decreasing), and consider the last non-free-fall segment $(\alpha^k, \tau^k, a^k)$, i.e., $k$ is the maximal $k$ for which $\alpha^k \neq 0$. Assume that $k > 1$ (if not, then $\pi$ is already a free-fall contract).

Let $\alpha_{BR} = \alpha_{a^k, a^k+1}$ be the indifference contract for the best-response boundary separating the current action from the previously incentivized action. Consider replacing this segment with the two consecutive segments $(\alpha_{BR}, (\alpha^k/\alpha_{BR})\tau^k, a^k)$, $(0, (1 - \alpha^k/\alpha_{BR})\tau^k, a^k))$. In doing so we essentially are doing the inverse of the first rewriting rule in Lemma B.2 – replacing a single segment with two segments that average to the original segment – and because of this, the resulting dynamic contract is valid and has the same utility as our original contract (the construction also guarantees both segments stay within this region). But now we have a segment $(\alpha_{BR}, (\alpha^k/\alpha_{BR})\tau^k, a^k)$ that lies along the best-response boundary $\alpha_{BR}$, so by Lemma B.3 we can replace it with the segment $(\alpha_{BR}, (\alpha^k/\alpha_{BR})\tau^k, a^k + 1)$ and strictly increase the utility of our dynamic contract (see Figure 5).

We can then merge this segment with the previous segment in (which also incentivizes action $a^k + 1$) to obtain a new dynamic contract with strictly greater utility than $\pi$ and whose first non-free-fall action occurs strictly earlier. Repeating this process, we obtain a free-fall contract $\pi'$ with at least the same utility as $\pi$. $\qquad\square$

We can now prove the main theorem of this section.

*Proof of Theorem 3.1.* From Lemmas B.4 and B.5 the first part of this theorem (that there exists a free-fall linear contract $\pi'$ with $\mathsf{Util}(\pi') \geq \mathsf{Util}(\pi)$) immediately follows.

To show that we can efficiently compute this free-fall contract, note that the optimal free-fall linear contract might as well start with a segment of the form $(\alpha_{i-1,i}, \tau, i)$ for some indifference contract $\alpha_{i-1,i}$ (if it does not start by offering some indifference contract, we can apply the rewriting rule of Lemma B.2 to merge this segment with the following segment, which would incentivize the same action).

It is also true that the optimal free-fall linear contract might as well *end* at an indifference contract: that is, $\overline{\alpha}^K = \alpha_{j-1,j}$ for some $j$. To see this, consider a free-fall linear contract $\pi$ that does not end on an indifference contract. It ends with a segment of the form $(0, \tau^K, a)$ for some agent action $a$. Consider the contract $\pi(\tau)$ formed by replacing the duration of the last segment with $\tau$; this operation is valid for all $\tau$ in some interval $[0, \tau_{\max}]$. Note that $\mathsf{Util}(\pi(\tau))$ is a convex function of $\tau$ (it is of the form $(\mathsf{Util}(\pi(0))\mathcal{T}^{K-1} + u_P(0,a)\tau)/(\mathcal{T}^{K-1} + \tau))$ so it is maximized when $\tau$ equals one of the endpoints of this interval. But at both endpoints, $\overline{\alpha}^K$ lies on a best-response boundary (for $\tau = 0$, $\alpha_{a,a+1}$, for $\tau = \tau_{\max}$, $\alpha_{a-1,a}$).

Since our optimal contract is completely characterized by its start and end points, it can be computed in polynomial time in $n$ by testing all the pairs of indifference points $\{\alpha_{i-1,i}, \alpha_{j-1,j}\}$ with $j \leq i$ as candidates for the start and end points of the optimal initial contract (this pair of indifference points also uniquely specifies the fraction of time that must be spent in free-fall). Note that in the case where in the optimal free-fall contract $i = j$, the optimal contract is the best static contract. $\qquad\square$

In Appendix D (see Theorem D.1), we generalize Theorem 3.1, showing that free-fall contracts are optimal for a much broader family of dynamic contracts with "single-dimensional scaling," where the principal is using an arbitrary non-linear contract and dynamically rescales it during the interaction with the agent.

Our proof of Theorem D.1 is parallel to the proof of Theorem 3.1 in the sense that we demonstrate how to gradually transform a general single-dimensional-scaling contract into a free-fall contract, while increasing utility for the principal. The main difficulty in applying the proof of Theorem 3.1 directly is that the rewriting rule in Lemma B.3 no longer holds – for general contracts with single-dimensional scaling, it is not the case that segments along a best-response boundary should always incentivize the higher action for the agent. In the proof of Theorem D.1, we forego the use of this rewriting rule and instead using the weaker condition that there cannot be two consecutive segments along a best-response boundary.

## C  Proof of Theorem 3.2 (Win-Win)

*Proof.* Consider the following contract game.[10] There are $n > 2$ actions, with expected reward $R_i = v^i$ for some $v > 0$. Concretely, we let $v = 2$. The cost of the action are specified recursively by $c_1 = 0$ and $c_i = c_{i-1} + R_{i-1} - \frac{1}{2}$ for $i > 1$, yielding $c_{i>1} = \sum_{k=2}^{i}(2^{k-1} - \frac{1}{2})$. The resulting indifference contracts are thus $\alpha_i = 1 - 2^{-i}$ for $1 < i \leq n$. In this game, the principal has the same utility (of one unit) for all the indifference contracts. The agent's utility under the contract $\alpha_i$, as the reader can verify, is given by $2^i - \frac{3}{2} - \sum_{k=1}^{i}(2^{k-1} - \frac{1}{2}) = \frac{1}{2}(1 + i)$. Notably, this utility is higher for the higher actions. The welfare of action $i$ is thus $w_i = \frac{1}{2}(3 + i)$. Next, we slightly alter this game by increasing the payoff of action 2 by a small amount $\epsilon > 0$ such that the optimal static contract is now $\alpha_2$, which yields a utility of $1 + \mathcal{O}(\epsilon)$ for the principal, and the agent is still indifferent under this contract between action 2 and the null action. In the following analysis, we are mainly interested in large (but finite) $n$. Notice that the optimal static contract is extremely inefficient for large $n$, getting an arbitrarily low (independent of $n$) fraction of the optimal welfare.

Now consider an optimal dynamic strategy; by Theorem 3.1, there is an optimal strategy of a free-fall form. We will construct a free-fall contract **p** that starts at $\alpha_n$, so action $n$ is played initially, where the duration $\lambda T$ of that stage is chosen such that the final action at time $T$ is action $\lceil \frac{1}{2}\log(n) \rceil$. Specifically, we require $\lambda \alpha_n + (1 - \lambda) \cdot 0 = \alpha_{\lceil \frac{1}{2}\log(n) \rceil}$, and so $\lambda = \frac{2^n}{2^n - 1}\left(1 - \frac{1}{\sqrt{n}}\right)$. We show that this free-fall strategy bounds the utilities of both players from below.

**Claim C.1.** *In an optimal free-fall contract, the utilities for both players are at least those obtained in the contract described above.*

For ease of presentation, the proof for this claim is deferred to the following subsection. It consists of three parts: first, we show that an optimal free-fall contract must start at $\alpha_n$. This is done directly by way of contradiction. Then, the proof shows that the last action that is played by the agent in an optimal free-fall contract must be higher than $\frac{1}{2}\log n$. The intuition for this part of the proof is that as the principal continues to free fall through lower and lower actions, the marginal gain from each action (which is the expected reward of that action because we are free falling) continues to diminish. At some point, the marginal gain is outweighed by the current average principal utility, which we show should occur at action $\Theta(\log n)$ (since we know the principal can get an average utility of $\Theta(n)$ and the expected reward of action $i$ is $2^i$). Lastly, we compare the utilities of both players in a free-fall strategy that begins at action $n$ and ends at action $\lceil \frac{1}{2}\log n \rceil$ to those of the optimal free-fall strategy and observe that the utilities in the former case bound the respective utilities in the latter case from below. The principal's utility is clearly bounded from below by her utility in our strategy due to optimality. For the agent, the total utility is determined by the stopping point. Since the agent's utility at $\alpha_i$ is increasing with $i$ in our game, we conclude that the agent's utility in an optimal contract is at least $\frac{1}{2}\log n$.

Now let us calculate the average utilities for the players under our dynamic strategy, averaged over the whole sequence of play. The agent's average utility at the last step is the same as the utility that would have been obtained under the average contract at that time, which is $\frac{1}{2}(1 + \lceil \frac{1}{2}\log(n) \rceil)$. To calculate the utility for the principal, we define $t_i$ to be the time when the agent switches from action $i$ to action $i - 1$. We know that the transition from action $n$ to $n - 1$ happens at time $t_n = \lambda T$, and until that time the principal gains a utility of one per time unit. After that time, the average contract at time $t$ is the weighted average until $t$ of the contract $\alpha_n$ with weight $\lambda T$ and zero contract with weight $t - \lambda T$. Therefore, the transition times from each action $i$ are given by $t_i = \lambda T \frac{\alpha_n}{\alpha_i}$. After time $\lambda T$, the principal pays zero and extracts the full welfare from the agents actions, and so the overall utility for the principal is $\lambda T + \sum_{i=\lceil \frac{1}{2}\log(n) \rceil}^{n}(t_{i-1} - t_i)R_{i-1}$.

**Claim C.2.** *The utility for the principal in the free-fall contract $(\alpha_n, \lambda)$ is $\mathcal{O}(n)$.*

---

[10]In this example we shift the rewards with an additive constant such that the reward for the principal when the agent takes the null action equals some constant instead of zero. This simplifies the following analysis and is without loss of generality.

*Proof.* The utility from region $i$ is $\lambda T + \sum_{i=\lceil \frac{1}{2}\log(n)\rceil}^{n} (t_{i-1} - t_i)R_{i-1}$. The time intervals are

$$(t_{i-1} - t_i) = \lambda T \alpha_n \Big(\frac{1}{\alpha_{i-1}} - \frac{1}{\alpha_i}\Big) = \frac{\lambda T \alpha_n 2^i}{(2^i - 2)(2^i - 1)}.$$

The utility for the principal from region $i > \frac{1}{2}\log(n)$ and large $n$ is thus:

$$\frac{\lambda T \alpha_n 2^{2i}}{2(2^i - 2)(2^i - 1)} = \Theta(1).$$

Summing over $n - \frac{1}{2}\log(n)$ such terms yields a utility of $\mathcal{O}(n)$. $\qquad\square$

The above arguments hold similarly also for perturbed versions of this game. For example, shifting the rewards by arbitrary and independent values in the range $[-1, 1]$, as well as re-scaling the reward parameter $v$, yielding a positive measure in the parameter space. $\qquad\square$

## C.1 Proof of Claim C.1

*Proof.* We execute this proof in two parts. In the first part of the proof, we will show that any optimal dynamic (free fall) contract must begin at $\alpha_n$. In the second part of the proof, we show that an optimal dynamic (free fall) contract that begins at $\alpha_n$ must end at $\alpha_{\lceil \frac{1}{2}\log n\rceil}$ or higher, if $n$ is sufficiently large. This is enough to imply the claim because if the optimal free fall contract stops at a higher action than $\lceil \frac{1}{2}\log n\rceil$, then the principal has higher utility due to optimality and the agent has higher utility since their utility is increasing in actions.

We now prove that any optimal dynamic contract must begin at $\alpha_n$. For the sake of contradiction, suppose that it instead begins at $\alpha_i$ for some action $i \in [1, n-1]$. In particular, it begins with the segment $(p^1 = \alpha_i R, \tau^1, a^1 = i)$ for some $i \in [1, n-1]$. To achieve a contradiction, we will show that this dynamic contract is not optimal by producing a better dynamic contract.

In particular, let us consider replacing this first segment with the following two segments: $(\alpha_{i+1}R, x \triangleq \frac{\alpha_i}{\alpha_{i+1}}\tau^1, i+1), (0, y \triangleq \big[1 - \frac{\alpha_i}{\alpha_{i+1}}\big]\tau^1, i)$ (and re-indexing all subsequent segments appropriately). We claim that this will achieve strictly greater principal utility, while leaving the total time unaffected. We first show how we solved for the appropriate time-split $(x, y)$.

$$x + y = \tau^1 \qquad ((x, y) \text{ is a time split})$$
$$x\alpha_{i+1} = \tau^1\alpha_i \qquad (\text{At time } \tau^1, \text{ the cumulative linear contract is still } \alpha_i)$$
$$x = \frac{\alpha_i}{\alpha_{i+1}}\tau^1$$
$$y = \Big[1 - \frac{\alpha_i}{\alpha_{i+1}}\Big]\tau^1$$

By construction, our choice of $x$ and $y$ keeps the total time invariant, so it remains to prove that this results in strictly more principal utility. Since all subsequent segments are the same and generate the same amount of principal utility, we only need to compare the principal utility of these three segments.

The (cumulative) principal utility of the original segment $(\alpha_i R, \tau^1, i)$ is just $\tau^1$ since the contract problem is designed so that all indifference contracts $\alpha_i$ result in one unit of utility to the principal. The exception is action one, which was adjusted to have $1 + O(\varepsilon)$ principal utility and therefore has cumulative principal utility $\tau^1(1 + O(\varepsilon))$.

Next, we consider the cumulative principal utility of our two new segments $(a_{i+1}R, x, i+1)$ and $(0, y, i)$. The first segment has (cumulative) principal utility equal to just $x$ for the same reason as above (but now $i+1$ cannot be the first action). The second segment has (cumulative) principal utility equal to $y(R_{i+1})$ where $R_{i+1}$ is the expected reward from action $i+1$, due to the fact that this segment offers the zero contract. Together, these two segments generate (cumulative) principal utility equal to the following.

$$x + y(R_{i+1}) = (x + y) + y(R_{i+1} - 1)$$
$$= \tau^1 + y(2^{i+1} - 1)$$

However, we can see from our choice of $y$ that $y > 0$ and $(2^{i+1} - 1) > 0$ since $i \geq 1$. Hence this strictly beats the cumulative principal utility of the original segment as long as $\varepsilon$ is sufficiently small. This completes our contradiction, since the original dynamic contract was assumed to be optimal but we found a strictly better one. Hence the optimal dynamic contract must free fall from $\alpha_n$ (which there is no higher action to start from instead), completing the first part of the proof.

We now use this fact to prove that the optimal dynamic (free fall) contract must end at $\alpha_{\lceil \frac{1}{2} \log n \rceil}$ or higher, if $n$ is sufficiently larger. The proof plan is to consider the effect of free falling through an additional action, and determining when that might improve the free fall contract. As a first step, we observe that the objective function of the continuous setting, $\mathsf{Util}(\pi)$, is invariant when we equally scale all times $\tau^k$. As a result, we can assume without loss of generality that the first segment of free-fall $(p^1 = \alpha_n R, \tau^1, a^1 = n)$ uses $\tau^1 = 1$. We can also assume without loss of generality that the other segments $\{(p^k = 0, \tau^k, a^k = n - k + 1)\}_{k=2}^{K}$ begin and end at region boundaries, which is enough to work out their durations $\tau^k$ based on when the average linear contract reaches a particular indifference point.

$$
\begin{aligned}
\tau^k &= \frac{\alpha_n}{\alpha_{n-k+1}} - \frac{\alpha_n}{\alpha_{n-k+2}} = \frac{1 - 2^{-n}}{1 - 2^{-n+k-1}} - \frac{1 - 2^{-n}}{1 - 2^{-n+k-2}} \\
&= [1 - 2^{-n}] \frac{2^{-n+k-1} - 2^{-n+k-2}}{(1 - 2^{-n+k-1})(1 - 2^{-n+k-2})} \\
&= [1 - 2^{-n}] \frac{2^{-n+k-2}}{(1 - 2^{-n+k-1})(1 - 2^{-n+k-2})}
\end{aligned}
$$

Hence segment $k \in [2, K]$ contributes the following (cumulative) principal utility.

$$
\begin{aligned}
\tau^k u_P(p^k, a^k) &= \tau^k 2^{n-k+1} = 2^{n-k+1} \cdot [1 - 2^{-n}] \frac{2^{-n+k-2}}{(1 - 2^{-n+k-1})(1 - 2^{-n+k-2})} \\
&= \frac{1}{2} [1 - 2^{-n}] \frac{1}{(1 - 2^{-n+k-1})(1 - 2^{-n+k-2})}
\end{aligned}
$$

Let $\pi_K$ be the trajectory that uses $K$ segments. We can compute its objective value to be the following.

$$
\begin{aligned}
\mathsf{Util}(\pi_K) &= \frac{1 + \frac{1}{2}[1 - 2^{-n}] \sum_{k=2}^{K} \frac{1}{(1-2^{-n+k-1})(1-2^{-n+k-2})}}{(1 - 2^{-n})/(1 - 2^{-n+K-1})} \\
&= (1 - 2^{-n+K-1}) \left[ 1/(1 - 2^{-n}) + \frac{1}{2} \sum_{k=2}^{K} \frac{1}{(1 - 2^{-n+k-1})(1 - 2^{-n+k-2})} \right]
\end{aligned}
$$

We can take the difference of two such expressions to decide whether $\pi_{K+1}$ is better than $\pi_K$. For $n - \left\lceil \frac{1}{2} \log n \right\rceil \leq K \leq n-1$:

$$
\begin{aligned}
\mathsf{Util}(\pi_{K+1}) - \mathsf{Util}(\pi_K) &= (1 - 2^{-n+K}) \left[ 1/(1-2^{-n}) + \frac{1}{2} \sum_{k=2}^{K+1} \frac{1}{(1-2^{-n+k-1})(1-2^{-n+k-2})} \right] \\
&\quad - (1 - 2^{-n+K-1}) \left[ 1/(1-2^{-n}) + \frac{1}{2} \sum_{k=2}^{K} \frac{1}{(1-2^{-n+k-1})(1-2^{-n+k-2})} \right] \\
&= (1 - 2^{-n+K}) \frac{1}{2} \frac{1}{(1-2^{-n+K})(1-2^{-n+K-1})} \\
&\quad - 2^{-n+K-1} \left[ 1/(1-2^{-n}) + \frac{1}{2} \sum_{k=2}^{K} \frac{1}{(1-2^{-n+k-1})(1-2^{-n+k-2})} \right] \\
&\leq \frac{1}{2} \frac{1}{(1-2^{-n+(n-1)})(1-2^{-n+(n-1)-1})} \\
&\quad - 2^{-n+(n-\frac{1}{2}\log n)-1} \left[ \frac{1}{2} \sum_{k=2}^{n-\left\lceil \frac{1}{2}\log n \right\rceil} 1 \right] \\
&= \frac{1}{2} \frac{1}{(1/2)(3/4)} - \frac{1}{2\sqrt{n}} \left[ \frac{1}{2} \left( n - \left\lceil \frac{1}{2} \log n \right\rceil - 1 \right) \right]
\end{aligned}
$$

Since the positive term has magnitude $O(1)$ and the negative term has magnitude $O(\sqrt{n})$, this bound will always be negative when $n$ is sufficiently large. Hence it is strictly not worth it to free fall below $\alpha_{\left\lceil \frac{1}{2} \log n \right\rceil}$, as desired. This completes the proof. $\qquad \square$

## D  General Contracts with Single-Dimensional Scaling

Here we consider general contracts, and in Theorem D.1 generalize the result of Theorem 3.1 to families of one-dimensional (yet non-linear) dynamic contracts for which free-fall contracts are optimal.

Given any contract $\mathbf{p}$, the set of $\mathbf{p}$-scaled contracts are the one-dimensional family of contracts of the form $\alpha \mathbf{p}$ for some $\alpha \geq 0$. We will consider a principal that is restricted to only play $\mathbf{p}$-scaled contracts. In the continuous-time formulation of Section 2.2, this means that each contract $p^k$ must be $\mathbf{p}$-scaled. We will let $p^k = \alpha^k \mathbf{p}$, and we will often abuse notation and write $\alpha^k$ as shorthand for this contract (e.g., we will specify segments of the trajectory $\pi$ in the form $(\alpha^k, \tau^k, a^k)$). Recall that a free-fall contract denotes such a dynamic contract for the principal where $\alpha^k = 0$ for all $k > 1$.

As with linear contracts, note that as $\alpha$ increases from 0, the contract $\alpha \mathbf{p}$ incentivizes the agent to play an action in $\mathsf{BR}_{\mathbf{p}}(\alpha)$ (which is unique except for at most $n$ "breakpoint" values of $\alpha$, where the agent is indifferent between two actions). This induces an ordering over the actions; we will relabel the actions so that actions 1 (the null action), 2, 3, ... are incentivized for increasing values of $\alpha$. Formally, if the agent has $n$ actions, we have $n$ "breakpoints" $0 = \alpha_{0,1} < \alpha_{1,2} < \alpha_{2,3} < \cdots < \alpha_{n-1,n}$, where action $i$ belongs to $\mathsf{BR}_{\mathbf{p}}(\alpha)$ iff $\alpha \in [\alpha_{i-1,i}, \alpha_{i,i+1}]$ (with $\alpha_{n,n+1} = \infty$).

Our main result in this section is the following theorem, by which free-fall $\mathbf{p}$-scaled contracts are optimal $\mathbf{p}$-scaled dynamic contracts.

**Theorem D.1.** *Let $\pi$ be any $\mathbf{p}$-scaled dynamic contract. Then there exists a free-fall $\mathbf{p}$-scaled contract $\pi'$ where $\mathsf{Util}(\pi') \geq \mathsf{Util}(\pi)$.*

To prove Theorem D.1, we will establish a sequence of lemmas constraining the potential geometry of an optimal $\mathbf{p}$-scaled dynamic contract. Note that since linear contracts are a specific case of $\mathbf{p}$-scaled contracts, this also provides an alternate proof of Theorem 3.1.

We begin our proof with the observation that, similar to linear contracts, $\mathbf{p}$-scaled contracts cannot "skip over" actions for the agent (c.f. Lemma B.1, which has an essentially identical proof).

**Lemma D.2.** *If $\pi = \{(\alpha^k, \tau^k, a^k)\}$ is a $\mathbf{p}$-scaled dynamic contract, then $\forall k, |a^k - a^{k+1}| \leq 1$.*

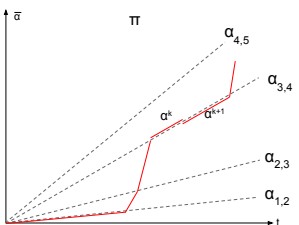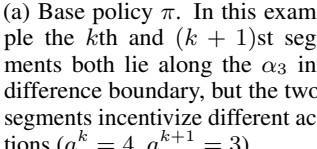

(a) Base policy $\pi$. In this example the $k$th and $(k+1)$st segments both lie along the $\alpha_3$ indifference boundary, but the two segments incentivize different actions ($a^k = 4$, $a^{k+1} = 3$).

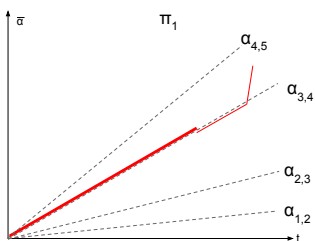

(b) Policy $\pi_1$ formed by replacing the first $k$ segments of $\pi$ with an enlarged version of the $k$th segment.

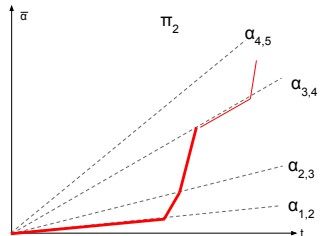

(c) Policy $\pi_2$ formed by replacing the first $k$ segments of $\pi$ with an enlarged version of the first $k-1$ segments.

Figure 6: Figures for the proof of Lemma D.3.

*Proof.* Note that $a^k$ and $a^{k+1}$ both must be best-responses to the average historical contract $\overline{\mathbf{p}}^k$ after segment $k$, which is a $\mathbf{p}$-scaled contract with parameter $\overline{\alpha}^k = \sum_{k'=1}^{k} \alpha^{k'} \tau^{k'} / \sum_{i=1}^{k} \tau^{k'}$. In other words, $a^k$ and $a^{k+1}$ belong to $\mathsf{BR}_{\mathbf{p}}(\overline{\alpha}^k)$. Since $\mathsf{BR}_{\mathbf{p}}(\alpha)$ is always of the form $\{i\}$ or $\{i, i+1\}$ the conclusion follows. $\square$

Now, recall that in Lemma B.2 we show that (for general contract problems) we can restrict our attention to trajectories where no two consecutive segments have the same agent best response (i.e., $a^k \neq a^{k+1}$ for any $k$). The following lemma proves a strengthening of this fact specific to $\mathbf{p}$-scaled contracts.

**Lemma D.3.** *Let $\pi$ be any $\mathbf{p}$-scaled dynamic contract. Then there exists a $\mathbf{p}$-scaled dynamic contract $\pi' = \{(\alpha^k, \tau^k, a^k)\}$ with the property that for all $k$, $a^k \neq a^{k+1}$ and $a^k \neq a^{k+2}$, and that $\mathsf{Util}(\pi') \geq \mathsf{Util}(\pi)$.*

*Proof.* The fact that we can rewrite $\pi$ into an equivalent contract where $a^k \neq a^{k+1}$ follows from the proof of Lemma B.2. Therefore, assume without loss of generality that $\pi$ already has this form. We will show how to rewrite it into a new dynamic contract $\pi'$ with the additional property that $a^k \neq a^{k+2}$.

We will induct on the number of segments in the path (it is obviously true when there is only $K = 1$ segment). Assume that for some $k$, $\pi$ has the property that $a^k = a^{k+2} \neq a^{k+1}$. This implies that $\mathsf{BR}_{\mathbf{p}}(\overline{\alpha}^k) = \mathsf{BR}_{\mathbf{p}}(\overline{\alpha}^{k+1}) = \{a^k, a^{k+1}\}$. Since there is a unique value of $\alpha$ for which $\mathsf{BR}_{\mathbf{p}}(\alpha) = \{a^k, a^{k+1}\}$ (namely, one of the breakpoints $\alpha_{i,i+1}$), this can only happen if $\overline{\alpha}^k = \overline{\alpha}^{k+1}$, which in turn means that $\alpha^{k+1} = \overline{\alpha}^k = \overline{\alpha}^{k+1}$. Pictorially, this is because if a dynamic contract spends only one segment in a best-response region, this segment must lie along the boundary of the best-response region (see Figure 6).

Now, consider the following two modifications of $\pi$:

1. In $\pi_1$, we replace the first $k+1$ segments of $\pi$ with a scaled up version of the $(k+1)$st segment. That is, remove the first $k+1$ segments of $\pi$ and replace them with $(\alpha^{k+1}, \mathcal{T}^{k+1}, a^{k+1})$. To see that this is a valid contract, note that since $\alpha^{k+1} = \overline{\alpha}^{k+1}$, $\alpha^{k+1}$ incentivizes action $a^{k+1}$ so the first segment of this contract is valid. Moreover, after $\mathcal{T}^{k+1}$ time units have elapsed, both $\pi$ and $\pi_1$ resume the same sequence of segments from the same state $\overline{\alpha}^{k+1}$.

2. In $\pi_2$, we replace the first $k+1$ segments of $\pi$ with a scaled up version of the first $k$ segments. That is, remove the segment $(\alpha^{k+1}, \tau^{k+1}, a^{k+1})$, and scale up $\tau^{k'}$ (for each $1 \leq k' \leq k$) to $\tau^{k'}(\mathcal{T}^{k+1}/\mathcal{T}^k)$. Again, this is a valid dynamic contract because scaling up a (prefix of a) dynamic contract results in a valid dynamic contract, and $\pi$ and $\pi_2$ both resume the remainder of segments at the same time and from the same state $\overline{\alpha}^{k+1}$.

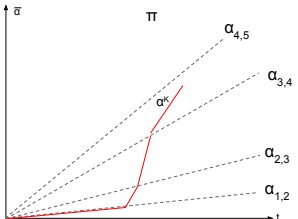

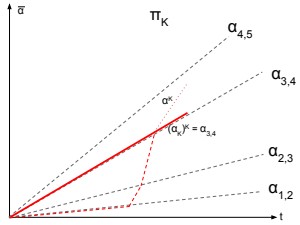

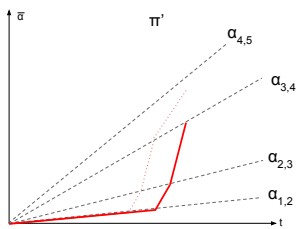

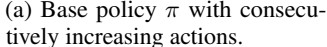

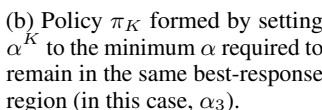

(a) Base policy $\pi$ with consecutively increasing actions.

(b) Policy $\pi_K$ formed by setting $\alpha^K$ to the minimum $\alpha$ required to remain in the same best-response region (in this case, $\alpha_3$).

(c) Policy $\pi'$ formed by scaling up the first $K - 1$ segments of $\pi$ (i.e., all but the last segment).

Figure 7: Figures for the proof of Lemma D.4.

Finally, note that $\mathsf{Util}(\pi)$ is a convex combination of $\mathsf{Util}(\pi_1)$ and $\mathsf{Util}(\pi_2)$ – specifically, $\mathsf{Util}(\pi) = (\tau^{k+1}\pi_1 + \mathcal{T}^k\pi_2)/\mathcal{T}^{k+1}$ – and so is less than or equal to one of them. But both $\pi_1$ and $\pi_2$ have strictly fewer segments than $\pi$, so by applying the inductive hypothesis, we are finished. $\qquad\square$

As a consequence of Lemmas D.2 and D.3, we can restrict ourselves to dynamic contracts whose sequences of actions are either consecutively increasing ($a^{k+1} = a^k + 1$) or consecutively decreasing ($a^{k+1} = a^k - 1$). We show that we can ignore the first case – such contracts can never be better than static contracts.

**Lemma D.4.** *Let $\pi = \{(\alpha^k, \tau^k, a^k)\}$ be a $\mathbf{p}$-scaled dynamic contract where the $a^k$ are consecutively increasing. Then there exists a static $\mathbf{p}$-scaled contract $\pi'$ (i.e., a single segment dynamic contract of the form $(\alpha', 1, a')$) where $\mathsf{Util}(\pi') \geq \mathsf{Util}(\pi)$.*

*Proof.* As in the proof of Lemma D.3, we will again induct on the number of segments of $\pi$. If $\pi$ has one segment, we are done.

Now consider a $\pi$ with $K$ segments, whose last segment is $(\alpha^K, \tau^K, a^K)$. Recall that for any $i$, $\alpha_{i-1,i}$ is the smallest value of $\alpha$ for which $\alpha\mathbf{p}$ incentivizes action $i$. Note that if $\alpha^K > \alpha_{a^K-1,a^K}$, we can improve the utility of the principal by decreasing $\alpha^K$ to $\alpha_{a^K-1,a^K}$ (this pays strictly less to the agent but still incentivizes the same action $a^K$). We'll therefore assume the last segment is of the form $(\alpha_{a^K-1,a^K}, \tau^K, a^K)$; note that this segment by itself is a valid static $\mathbf{p}$-scaled contract, as $\alpha_{a^K-1,a^K}$ incentivizes action $a^K$. Call this contract $\pi_K$.

Let $\pi'$ be the dynamic contract formed by the first $K - 1$ segments of $\pi$ (see Figure 7 for examples of $\pi_K$ and $\pi'$). But now, $\mathsf{Util}(\pi)$ is a convex combination of $\mathsf{Util}(\pi')$ and $\mathsf{Util}(\pi^K)$, so it is at most the maximum of these two quantities. If this maximum is $\mathsf{Util}(\pi^K)$, we are done ($\pi^K$ is a static contract); if it is $\mathsf{Util}(\pi')$, we are also done by the inductive hypothesis ($\pi'$ has $K - 1$ segments). $\qquad\square$

Finally, we show that in the case where the sequence of actions are consecutively decreasing, such a contract is no better than some free-fall contract.

**Lemma D.5.** *Let $\pi = \{(\alpha^k, \tau^k, a^k)\}$ be a $\mathbf{p}$-scaled dynamic contract where the $a^k$ are consecutively decreasing. Then there exists a free-fall $\mathbf{p}$-scaled contract $\pi'$ where $\mathsf{Util}(\pi') \geq \mathsf{Util}(\pi)$.*

*Proof.* Assume that $\pi$ is not a free-fall contract. We will show we can rewrite $\pi$ in a way so that either the first agent action $a^1$ strictly decreases or the first non-free-fall occurs strictly later. Since the number of segments in $\pi$ is bounded (by $n$, since the actions are consecutively decreasing), this implies the theorem statement.

Let $(\alpha^k, \tau^k, a^k)$ be the first segment in $\pi$ with $k \geq 2$ where $\alpha^k > 0$ (so, the dynamic contract is not free-falling here). Note that since this segment ends on the boundary between the best-response regions for $a^k$ and $a^{k+1} = a^k - 1$, $\overline{\alpha}^k = \alpha_{a^k-1,a^k}$.

The main observation of this proof is that we can rewrite this segment as a combination of a free-fall segment (with $\alpha = 0$) and a segment along the boundary of these two best-response regions (with

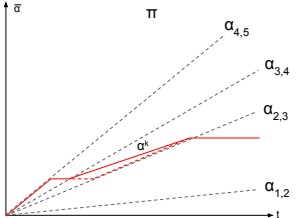
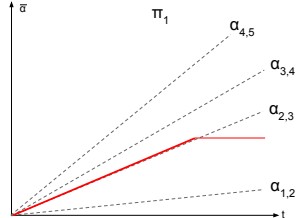
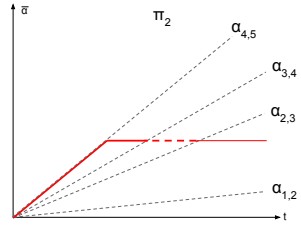

(a) Base policy $\pi$ with consecutively decreasing actions. The $k$th segment is the first non-free-fall segment: we form $\pi'$ (the dashed trajectory) by decomposing this segment into a free-fall segment followed by a boundary segment.

(b) Policy $\pi_1$ formed by replacing the first $k$ segments of $\pi$ by a scaled up version of the boundary dashed segment.

(c) Policy $\pi_2$ formed by replacing the first $k$ segments of $\pi$ by a scaled up version of the prefix of $\pi$ up to (but not including) the boundary dashed segment.

Figure 8: Figures for the proof of Lemma D.5.

$\alpha = \overline{\alpha}^k$). Specifically, form a new dynamic contract $\pi'$ by replacing $(\alpha^k, \tau^k, a^k)$ in $\pi$ with the two consecutive segments $(0, \lambda\tau^k, a^k)$ and $(\alpha_{a^k-1, a^k}, (1 - \lambda)\tau^k, a^k)$, where $\lambda$ is chosen so that $(1 - \lambda)\alpha_{a^k-1, a^k} = \alpha^k$. Note that by doing this $\pi'$ now has $K + 1$ segments, where segments $k$ and $k + 1$ are this new free-fall and boundary segment respectively. Note that we will let quantities like $\tau^k$, $\mathcal{T}^k$, and $\overline{\alpha}^k$ still refer to the relevant quantities for $\pi$, not $\pi'$.

This allows us to proceed via a similar technique as in Lemma D.3. Consider the following two modifications of $\pi'$ (see Figure 8 for examples):

1. In $\pi_1$, we replace the first $k + 1$ segments of $\pi'$ with a scaled-up version of the boundary segment of the form $(\overline{\alpha}^k, \mathcal{T}^k, a^k)$.

2. In $\pi_2$, we replace the first $k + 1$ segments of $\pi'$ with a scaled up version of the first $k$ segments (the first $k - 1$ segments of $\pi$ and the free-fall segment, but not the boundary segment). Specifically, let $C = \mathcal{T}^k/(\mathcal{T}^{k-1} + \lambda\tau^k)$. Then the first $k - 1$ segments of $\pi_2$ are of the form $(\alpha^{k'}, C\tau^{k'}, a^{k'})$, and the $k$th segment of $\pi_2$ is of the form $(0, C\lambda\tau^k, a^k)$.

As in the proof of Lemma D.3, we can check that both $\pi_1$ and $\pi_2$ are valid dynamic contracts: in particular, after $\mathcal{T}^k$ units of time, they are both in the state $\overline{\alpha}^k$, so the remaining suffix of $\pi$ is a valid extension for both contracts.

Again, $\mathsf{Util}(\pi)$ can be written as a convex combination of $\mathsf{Util}(\pi_1)$ and $\mathsf{Util}(\pi_2)$, specifically,

$$\mathsf{Util}(\pi) = \frac{(1 - \lambda)\tau^k\mathsf{Util}(\pi_1) + (\mathcal{T}^{k-1} + \lambda\tau^k)\mathsf{Util}(\pi_2)}{\mathcal{T}^k}.$$

But $\pi_1$ starts at a later action than $\pi$ (since $a^k = a^1 - (k - 1)$), and $\pi_2$ is a free-fall contract for one further step than $\pi$ (since $\alpha^2 = \alpha^3 = \cdots = \alpha^{k-1} = 0$, and the $k$th segment in $\pi_2$ also has $\alpha = 0$). This completes the proof. $\square$

We can now conclude the proof of Theorem D.1.

*Proof of Theorem D.1.* Because of Lemmas D.2 and D.3, we can assume without loss of generality that the actions $a^k$ in $\pi$ are either consecutively increasing or decreasing. The conclusion now immediately follows from Lemmas D.4 and D.5. $\square$

# E   Proof of Thorem 4.2 (Unknown Time Horizon)

*Proof.* Due to Theorem H.3, it suffices to show that there exists some $\underline{\gamma}$ such that $U_{\underline{\gamma}}^\star < (1 + \varepsilon)R_\star$. This lets us focus on the continuous-time setting.

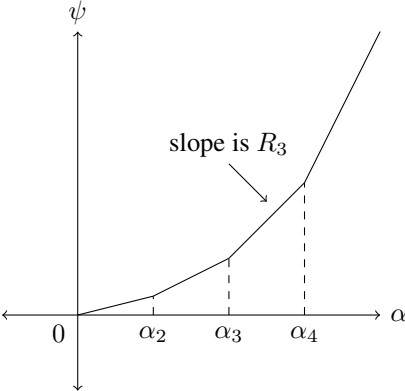

Figure 9: We use a "raw" potential function $\psi(\alpha)$ which maps time-averaged linear contracts $\alpha$ to (raw) potentials.

The high-level plan from here is to focus on a particular continuous trajectory $\pi = \left\{ (p^k, \tau^k, a^k) \right\}_k$ apply a potential argument to it. We will then show our analysis extends to distributions $\mathcal{D}$ for free. We will define a potential function $\psi(\alpha)$ that maps time-averaged linear contracts $\alpha$ to potentials in $\mathbb{R}_{\geq 0}$. This potential is based only on the principal-agent problem $(c, F, r)$. There are some peculiarities about our potential argument, relating to the passage of time. Consider a principal managing to produce a time-averaged linear contract of $\alpha$ after $t$ units of time, and compare that with a principal that has managed to arrive a time-averaged linear contract of $\alpha$ after $2t$ units of time instead, i.e. twice the time. In terms of absolute (not time-averaged) units of profit we can extract from this point, it is twice as good to be in the latter situation. With this in mind, our proof will carefully distinguish between the *raw potential* $\psi(\alpha)$ and the *time-weighted potential* $\psi(\alpha) \cdot t$. If a principal maintains a steady time-averaged linear contract, then the raw potential will remain constant while the time-weighted potential will grow.

The purpose of the time-weighted potential is to model the ability of a principal to extract additional profit by gradually lowering time-averaged linear contract. It will be used to demonstrate that this extra profit produced by using up a finite resource, which will imply the desired theorem result.

We now give our raw potential function $\psi(\alpha)$. We begin by writing down the linear contract breakpoints of $(c, F, r)$; without loss of generality[11] they are $0 < \alpha_2 < \alpha_3 < \cdots < \alpha_n$, where the linear contract $\alpha_i$ leaves the agent indifferent between actions $i-1$ and $i$. For notational convenience, we also define an $\alpha_1 \triangleq 0$ as the minimum linear contract to incentivize the first action. We also denote the expected reward of action $i$ with $R_i$. With this notation in place, our raw potential function $\psi : [0, \alpha_n] \to \mathbb{R}_{\geq 0}$ is the following piecewise-linear function. Note that we can assume without loss of generality that the average linear contract never exceeds $\alpha_n$, because capping it to this quantity only improves principal utility at all moments in time.

$$\psi(\alpha) \triangleq \begin{cases} \sum_{i=1}^{i'-1}(\alpha_{i+1} - \alpha_i)R_i + (\alpha - \alpha_{i'})R_{i'} & \text{if } \alpha \in [\alpha_{i'}, \alpha_{i'+1}) \\ \sum_{i=1}^{n-1}(\alpha_{i+1} - \alpha_i)R_i & \text{if } \alpha = \alpha_n \end{cases}$$

The potential above is depicted in Figure 9 and can be seen as the product of the following thought experiment: what if the principal was allowed to offer unbounded payments (in particular, payments can be negative and can exceed the payment bound $P$)? In our continuous-time setting, this gives the principal the ability to produce segments of play $(p^k, \tau^k, a^k)$ which have near-instantaneous times $\tau^k \to 0$ while using large-magnitude cumulative contracts $p^k \tau^k$ to move between the boundaries between actions. If these near-instantaneous actions are used at time $t$, then the time-weighted potential $\psi(\alpha) \cdot t$ captures the necessary payments to alter the time-averaged linear contract. One interesting aside about this thought experiment is that the necessary payment to near-instantaneously move up from $\alpha_i$ to the next $\alpha_{i+1}$, namely $[\psi(\alpha_{i+1}) - \psi(\alpha_i)] t$, is equal to the payout received for near-instantaneously using a negative contract to move down from $\alpha_{i+1}$ to $\alpha_i$.

---

[11] Implicitly, this step prunes away all actions which cannot be incentivized by a linear contract.

Potential function in hand, we return to the original problem where payments are bounded and nonnegative. Let us consider the $k^{th}$ segment of play ($p^k = \alpha^k R, \tau^k, a^k$) and relate the total profit generated during this segment of play with the change in potential.

For notational convenience we define shorthand for the cumulative linear contract offered.

$$\mathcal{A}^k \triangleq \sum_{k'=1}^{k} \tau^{k'} \alpha^k$$

We will also use $u_P^k$ to denote the (time-weighted) principal utility for segment $k$ and $u_\star^k$ to denote the corresponding amount of principal utility that the optimal static contract obtains over $\tau^k$ time. Using this notation, we can compute an upper bound on how much additional principal utility this segment manages to achieve over the optimal static contract.

$$u_P^k = \left[ (1 - \alpha^k) R_{a^k} \right] \tau^k$$

$$u_\star^k = \left[ \max_a (1 - \alpha_a) R_a \right] \tau^k$$

$$\geq \left[ (1 - \alpha_{a^k}) R_{a^k} \right] \tau^k$$

$$(u_p^k - u_\star^k) \leq \left[ (\alpha_{a^k} - \alpha^k) R_{a^k} \right] \tau^k$$

At the same time, this contract has shifted the time-averaged linear contract and hence altered the time-weighted potential.

$$\psi \left( \mathcal{A}^k / \mathcal{T}^k \right) \mathcal{T}^k - \psi \left( \mathcal{A}^{k-1} / \mathcal{T}^{k-1} \right) \mathcal{T}^{k-1}$$

$$= \left[ \sum_{i=1}^{\alpha^k} (\alpha_i - \alpha_{i-1}) R_{i-1} + \left( \mathcal{A}^k / \mathcal{T}^k - \alpha_{a^k} \right) R_{a^k} \right] \mathcal{T}^k$$

$$- \left[ \sum_{i=1}^{\alpha^k} (\alpha_i - \alpha_{i-1}) R_{i-1} + \left( \mathcal{A}^{k-1} / \mathcal{T}^{k-1} - \alpha_{a^k} \right) R_{a^k} \right] \mathcal{T}^{k-1}$$

$$= \left[ \mathcal{T}^k \sum_{i=1}^{\alpha^k} (\alpha_i - \alpha_{i-1}) R_{i-1} + \left( \mathcal{A}^k - \mathcal{T}^k \alpha_{a^k} \right) R_{a^k} \right]$$

$$- \left[ \mathcal{T}^{k-1} \sum_{i=1}^{\alpha^k} (\alpha_i - \alpha_{i-1}) R_{i-1} + \left( \mathcal{A}^{k-1} - \mathcal{T}^{k-1} \alpha_{a^k} \right) R_{a^k} \right]$$

$$= \tau^k \sum_{i=1}^{\alpha^k} (\alpha_i - \alpha_{i-1}) R_{i-1} + \left[ \left( \alpha^k \tau^k - \tau^k \alpha_{a^k} \right) R_{a^k} \right]$$

Interestingly, the expression for time-weighted potential has a term that perfectly cancels with our bound for how much additional principal utility this segment produces over the optimal static contract.

$$(u_p^k - u_\star^k) + \psi \left( \mathcal{A}^k / \mathcal{T}^k \right) \mathcal{T}^k - \psi \left( \mathcal{A}^{k-1} / \mathcal{T}^{k-1} \right) \mathcal{T}^{k-1} \leq \tau^k \sum_{i=1}^{\alpha^k} (\alpha_i - \alpha_{i-1}) R_{i-1}$$

$$\leq \int_{\mathcal{T}^{k-1}}^{\mathcal{T}^k} \psi \left( \frac{\mathcal{A}^{k-1} + (\mathcal{T} - \mathcal{T}^{k-1}) \alpha^k}{\mathcal{T}} \right) d\mathcal{T}$$

The right-hand side expression above is just the integral of the current raw potential as this segment advances the time from $\mathcal{T}^{k-1}$ to $\mathcal{T}^k$. Conveniently, this upper bound still works out to the same amount even if we subdivide our segment $(p^k, \tau^k, a^k)$ into two sub-segments $(p^k, x, a^k), (p^k, y, a^k)$ such that $x, y \in [0, \tau^k]$ and $x + y = \tau^k$ (and re-index the other segments appropriately). This means we can sum this bound to get an overall bound for any time $t \in [0, \overline{T}]$, just by splitting the last

segment appropriately. To formalize this, we introduce some more parenthetical superscript notation to denote the corresponding objects when considering time from zero to $t$. In particular, $u_\star^{(t)}$ denotes the optimal static contract's principal utility for $t$ units of time, $\mathcal{A}^{(t)}$ denotes the cumulative linear contract for $t$ units of time.

$$(u_p^{(t)}(\pi) - u_\star^{(t)}) + \psi\left(\mathcal{A}^{(t)}/t\right)t \leq \int_0^t \psi\left(\frac{\mathcal{A}^{(\mathcal{T})}}{\mathcal{T}}\right) d\mathcal{T}$$

Recall our notation where $R_\star$ denotes the optimal static contract's principal utility. For $t \in [\underline{T}, \overline{T}]$, we know that the excess principal utility needs to be at least $\varepsilon R_\star t$, which implies the following.

$$\varepsilon R_\star t + \psi\left(\mathcal{A}^{(t)}/t\right)t \leq \int_0^t \psi\left(\frac{\mathcal{A}^{(\mathcal{T})}}{\mathcal{T}}\right) d\mathcal{T}$$

$$\psi\left(\mathcal{A}^{(t)}/t\right) \leq -\varepsilon R_\star + \frac{1}{t}\int_0^t \psi\left(\frac{\mathcal{A}^{(\mathcal{T})}}{\mathcal{T}}\right) d\mathcal{T}$$

With this bound in mind, we can view every trajectory $\pi$ that manages to successfully beat the optimal static contract by $(1 + \varepsilon)$ in terms of how much raw potential it has as a function of time. Note that this bound controls the current raw potential based on the average raw potential up to this point (minus a constant). As a result, if we just consider trajectories $\pi$ that obey this bound, the worst case for us would be a function that satisfies it with equality everywhere since greedily picking the maximum value for the function early on allow for higher values later on (greedy stays ahead). We now solve for this function $f(t)$ which simultaneously maximizes raw potential everywhere.

$$\varepsilon R_\star t + f(t)t = \int_0^t f(\mathcal{T})d\mathcal{T}$$
$$\varepsilon R_\star + f(t) + f'(t)t = f(t)$$
$$f'(t) = -\varepsilon R_\star/t$$

At time $\underline{T}$, we know the raw potential can be at most $\psi(\alpha_n)$. We want to choose $\underline{\gamma}$ and hence $\overline{T}$ so that $f(\overline{T})$ is negative in order to create a contradiction. Because $f$ yields the maximum possible function value attainable at time $\overline{T}$, this means that our actual raw potential will also be negative at $\overline{T}$. We now solve for the largest value of $\gamma$ that does not actually create a contradiction.

$$f(\overline{T}) - f(\underline{T}) = -\psi(\alpha_n)$$
$$\int_{\underline{T}}^{\overline{T}} f'(t)dt = -\psi(\alpha_n)$$
$$-\varepsilon R_\star [\ln t]_{\underline{T}}^{\overline{T}} = -\psi(\alpha_n)$$
$$\ln(\overline{T}/\underline{T}) = \frac{\psi(\alpha_n)}{\varepsilon R_\star}$$
$$\gamma = e^{\phi(\alpha_n)/(\varepsilon R_\star)}$$

Hence it suffices to pick a $\underline{\gamma} > e^{\phi(\alpha_n)/(\varepsilon R_\star)}$. This demonstrates that it is impossible for a *deterministic* trajectory $\pi$ to beat the optimal static contract by a $(1 + \varepsilon)$ multiplicative factor.

What about *randomized* dynamic contracts $\mathcal{D}$? We can just take the appropriate convex combination of our bounds according to drawing $\pi \sim \mathcal{D}$. In particular, this yields:

$$\mathbb{E}_{\pi\sim\mathcal{D}}\left[(u_p^{(t)}(\pi) - u_\star^{(t)})\right] + \mathbb{E}_{\pi\sim\mathcal{D}}\left[\psi\left(\mathcal{A}^{(t)}/t\right)\right]t \leq \int_0^t \mathbb{E}_{\pi\sim\mathcal{D}}\left[\psi\left(\frac{\mathcal{A}^{(\mathcal{T})}}{\mathcal{T}}\right)\right] d\mathcal{T}$$

We can then re-execute the remainder of the proof in the same way, replacing the deterministic additional principal utility with expected additional principal utility and deterministic raw potential with expected raw potential. The expected potential function is still bounded everywhere by the same function $f(T)$ and we reach the same conclusions about $\underline{\gamma}$. This completes the proof. $\qquad\square$

**Remark.** *Due to Yao's minimax principle, Theorem 4.2 implies that there exists an adversarial distribution over times in $[\underline{T}, \overline{T}]$ such that for any randomized principal strategy, the ratio between expected principal utility and the principal utility of the optimal static contract for that duration of time is strictly less than $(1+\varepsilon)$. In order to apply Yao's minimax principle, we need the set of relevant principal strategies and the set of relevant adversary strategies to be finite. We already do this in our proof of Theorem H.3: the latter can just be an $\varepsilon$-net since principal utility is Lipschitz with Lipschitz constant depending on the contract problem, and after that the former then follows from Carathéodory's Theorem by treating each deterministic trajectory as a vector with one coordinate for every point in our $\varepsilon$-net.*

## F    Proof of Theorem 4.3 (Unknown Time Horizon – Converse)

*Proof.* We prove this by proving the contrapositive. Suppose for any fixed time $T$ there is a dynamic contract that can achieve an expected utility of $(1 + \epsilon)u_\star T$ for some $\epsilon > 0$. By Theorem 3.1, we can assume without loss of generality that this is a free-fall linear contract. We will show that for any $\gamma$ we will construct a dynamic contract such that for all $\underline{T} \in \mathbb{R}$ and all $t \in [\underline{T}, \gamma \cdot \underline{T}]$, we can achieve an expected utility of $(1 + f(\epsilon, \gamma)) \cdot u_\star \cdot t$ where $f(\epsilon, \gamma) \geq \Omega\left(\min\left(\left(\frac{\varepsilon}{4}\right)^{O(\log(1+\gamma))}, \frac{\varepsilon}{\gamma}\right)\right)$.

As a first step, we will show that if there is a free-fall linear contract that beats the optimal static contract, then there is a free-fall linear contract that beats the optimal static contract but also either (1) ends at or above the optimal static contract or (2) begins at the optimal static contract. Afterwards, we plan to analyze case (1) and (2) separately.

If our free-fall linear contract does not already satisfy case (1) or (2), then it must do one of the following; (a) begin at a higher breakpoint than the optimal static contract and end at a lower breakpoint than the optimal static contract or (b) being and end at lower breakpoints than the optimal static contract. We now analyze these two cases. In the process, we will lose a constant factor which is folded into our $\Omega$ notation.

**Case A: Dynamic contracts beginning above $\alpha_\star$ and ending below $\alpha_\star$.** We write our free-fall linear contract in the usual form $\pi = \{(\mathbf{p}^k, \tau^k, a^k)\}_{k=1}^K$. By virtue of being in this case, we know there is some index $2 \leq i < K$ such that the average linear contract after $i$ segments, $\overline{\mathbf{p}}^i$, is exactly $\alpha_\star$. We "cut" the trajectory $\pi$ at this point to produce two new trajectories $\pi'$ and $\pi''$. Specifically, $\pi' = \{(\mathbf{p}^k, \tau^k, a^k)\}_{k=1}^i$ and $\pi'' = \{(\alpha_\star, \mathcal{T}^i, a^i)\} \circ \{(\mathbf{p}^k, \tau^k, a^k)\}_{k=i+1}^K$ where $\circ$ denotes concatenation. In other words, we construct $\pi'$ by ending at this point and we construct $\pi''$ by taking the optimal static contract to this point and continuing as normal. Observe that the combined performance of $\pi'$ and $\pi''$ is equal to the combined performance of $\pi$ and just playing the single segment $\{(\alpha_\star, \mathcal{T}^i, a^i)\}$: $(1+\epsilon)u_\star \mathcal{T}^K + u_\star \mathcal{T}^i$. This results in a combined time-averaged performance of

$$\frac{(1 + \epsilon)u_\star \mathcal{T}^K + u_\star \mathcal{T}^i}{\mathcal{T}^K + \mathcal{T}^i} = u_\star \left[(1 + \epsilon)\frac{\mathcal{T}^K}{\mathcal{T}^K + \mathcal{T}^i} + (1)\frac{\mathcal{T}^i}{\mathcal{T}^K + \mathcal{T}^i}\right]$$
$$\geq (1 + \epsilon/2)u_\star$$

since $\mathcal{T}^K \geq \mathcal{T}^i$. Since $\pi'$ and $\pi''$ have this combined average, one of them must have at least this average (and we only lost a factor $1/2$ on our $\epsilon$, which is indeed a constant. Since $\pi'$ matches case (1) and $\pi''$ matches case (2), this completes the analysis of case (a).

**Case B: Dynamic contracts beginning and ending below $\alpha_\star$.** We take the obvious approach and choose to begin at $\alpha_\star$ instead. Specifically, we replace the first segment with a sequence of segments that begins at $\alpha_\star$ and then undergoes the appropriate number of free-fall segments to arrive at the same endpoint as before (same total time and average linear contract). We argue that each new segment has at least as much principal utility per unit time as the original segment. Since the total time is the same, this is a direct improvement over the original dynamic contract, both in terms of total principal utility and time-averaged principal utility. The argument that each new segment does at least as well per unit time is similar to before. The first new segment just hovers at the optimal static contract, which by definition is better than any other static contract (which our original segment must be). The remaining new segments are freefall segments, and achieve principal utility per unit time equal to the expected revenue of the actions they fall through. We observe that we fall through segments in order of decreasing expected utility, meaning all of these segments have higher expected utility than the action we originally began with, and expected revenue is at least the principal utility

of the static contract that achieves a particular action. We finish this case by noting that we did not diminish $\epsilon$ at all, which trivially a constant factor.

This completes our analysis of cases (a) and (b). In all cases, we managed to reduce to either case (1) or (2), which we now consider.

**Case 1: Dynamic contracts ending at or above $\alpha_\star$.** First, we consider the case where for any fixed $T$ there is a dynamic contract $\pi(T) = ((\alpha^1, \tau^1, a^1), \ldots, (\alpha^k, \tau^k, a^k))$ which ends at or above the optimal static action: $a^k \geq a_\star$. Given any $\gamma$ and time period $[\underline{T}, \overline{T} = \gamma \cdot \underline{T}]$, consider the dynamic contract which starts with $\pi(\underline{T})$, free falls to the optimal static contract, and then plays the optimal static contract for the remaining time period. We again observe (as we did for case (b)) that free falling through actions that are at least the $a_\star$ results in at least $u_\star$ principal profit per unit time. Hence the total revenue for any time $t \in [\underline{T}, \overline{T}]$ for the principal is $(1 + \epsilon) \cdot u_\star \underline{T} + (t - \underline{T}) \cdot u_\star$, which is at least $(1 + \epsilon/\gamma) R_\star t$.

**Case 2: Dynamic contracts starting in $\alpha_\star$.** By Theorem 3.1, we know any dynamic contract can be transformed into a free-fall dynamic contract with no loss in revenue. Therefore, we assume that for any fixed time horizon $T$, there is a dynamic contract form $\pi(T) = \big( \alpha_\star, \tau^1, a^1), (0, \tau^2, a^2), \ldots, (0, \tau^k, a^k) \big)$ which achieves a total revenue of $(1 + \varepsilon) R_\star T$. Since this is a free-fall contract, the optimal revenue from this contract can be characterized as $(1 - \alpha_\star) R_\star \tau^1 + \sum_{i=2}^{k} \tau^i R_{a_i}$ which is at least $(1 + \varepsilon) R_\star t > (1 + \varepsilon)(1 - \alpha_\star) R_\star$. Let $\mu$ be the minimum fraction of time such that for any time $T$, the dynamic contract $\pi(T)$ achieves revenue at least $(1 + \varepsilon/2) \mu u_\star T$. Since we know that $\pi(T)$ achieves a total revenue of $(1 + \varepsilon) u_\star T$ and starts out at the optimal static contract, we know that $\mu \geq \tau^1 / \sum_{i=1}^{k} \tau^i$ and it is a constant bounded away from 1. Let $S_i = \lceil \mu^i \overline{T} \rceil$ and let $p$ be the first index where $S_p$ is less than $\underline{T}$ (i.e., $p = \lceil \frac{\log(1+\gamma)}{\log(\mu)} \rceil$). By construction, $S_i$ satisfy two properties:

1. $S_p \leq \underline{T} \leq S_{p-1} \leq \ldots S_1 \leq \overline{T}$.

2. If the principal runs dynamic contract ending at $S_i$, namely $\pi(S_i)$, then they are guaranteed revenue $(1 + \varepsilon/2) t R_\star$ for any $t \in [S_{i+1}, S_i]$.

We will construct a sequence of dynamic contracts $\pi^i$ which have the property that for any $t \in [S_i, \gamma \underline{T}]$ achieves revenue that is at least $(1 + (\varepsilon/4)^i) R_\star t$. We do this via induction. For the base case, let $\pi^1 = \pi(\overline{T})$. By construction, we know that for all $t \in [S_1, \overline{T}]$, the principal will get revenue $(1 + \varepsilon/2) u_\star t$. Now suppose we have such a dynamic contract $\pi^i$, then we construct $\pi^{i+1}$ by taking a convex combination of $\pi^i$ and the optimal dynamic contract ending at $\pi(S_i)$. In particular, let

$$\pi^{i+1} = \frac{1 + \varepsilon/2}{1 + \varepsilon/2 + (\varepsilon/4)^i} \pi^i + \frac{(\varepsilon/4)^i}{1 + \varepsilon + (\varepsilon/4)^i} \pi(S_i).$$

For any $t \in [S_i, \overline{T}]$, we have that revenue we attain is at least the revenue from the contract

$$\frac{1 + \varepsilon/2}{1 + \varepsilon/2 + (\varepsilon/4)^i} \text{Revenue}(\pi^i(t)) \geq \frac{1 + \varepsilon/2}{1 + \varepsilon/2 + (\varepsilon/4)^i} (1 + (\varepsilon/4)^i) u_\star t \geq$$

$$1 + \frac{\varepsilon^{i+1}/2 \cdot 4^i}{1 + \varepsilon/2 + (\varepsilon/4)^i} \geq (1 + (\varepsilon/4)^{i+1}) u_\star t.$$

For any $t \in [S_{i+1}, S_i]$, observe that we get at least $u_\star t$ from the first contract $\pi^i$ and at least $(1 + \varepsilon/2) u_\star t$ in the second contract. Therefore we get at least

$$\frac{1 + \varepsilon/2}{1 + \varepsilon/2 + (\varepsilon/4)^i} u_\star t + \frac{(\varepsilon/4)^i}{1 + \varepsilon/2 + (\varepsilon/4)^i} (1 + (\varepsilon/2)) u_\star t \geq (1 + (\varepsilon/4)^i) u_\star t.$$

$\square$

## G  General Contracts

In this section, we give a general contract instance with $n = 4$ actions (3 non-null actions) and $m = 4$ outcomes (3 non-null outcomes), where the best dynamic contract provably outperforms the best free-fall dynamic contract. The instance in question is defined as follows:

- The cost vector $c = (c_1, c_2, c_3, c_4) = (0, 0.2, 0.4, 0.5)$.
- The reward vector $r = (r_1, r_2, r_3) = (0, 1.0, 1.6, 2.0)$.
- The forecast matrix is given by $F = \begin{pmatrix} 1.00 & 0.00 & 0.00 & 0.00 \\ 0.45 & 0.20 & 0.25 & 0.10 \\ 0.35 & 0.05 & 0.25 & 0.35 \\ 0.15 & 0.30 & 0.30 & 0.25 \end{pmatrix}$

This instance was found by a programmatic search[12] over a large collection of instances. For this instance, we can (again, programmatically) compute that the best free-fall dynamic contract achieves a net asymptotic utility for the principal of at most $0.753$ per round. At the same time, we can exhibit a non-free-fall dynamic contract for this instance that achieves a utility of at least $0.764$ per round. For conciseness, we present the details of our approach in Appendix G.1, where we construct well-tailored linear programs that provide the aforementioned intricate instance.

### G.1 Programmatic LP Search for Sub-Optimal Free Fall Against Non-Linear Contracts

At a high level, the verification of the example of section 3.3 relies on the following fact: given a sequence of actions $(a^1, a^2, \ldots, a^K)$, we can construct a polynomial-sized linear program to find the optimal continuous-time dynamic (general or free-fall) contract $\{(\mathbf{p}^k, \tau^k, a^k)\}_{k=1}^K$ with this specific action sequence.

The variables of this LP are the $\tau^k$ and $\mathbf{p}^k$ corresponding to each action $a^k$. The constraints follow from the definition of a valid trajectory of play in Section 2.2 and are as follows:

- **(Non-negativity)** $\mathbf{p}^k, \tau^k \geq 0$.

- **(Time normalization)** $\sum_{k=1}^K \tau^k = 1$. We normalize the total duration of the trajectory to 1.

- **(Beginning of segment is best response)** $\sum_{k'=1}^{k-1} \tau^{k'} u_L\left(\mathbf{p}^{k'}, a^k\right) \geq \sum_{k'=1}^{k-1} \tau^{k'} u_L\left(\mathbf{p}^{k'}, a'\right)$ for any $a' \in [n]$. This represents the constraint $a^k \in \mathsf{BR}(\overline{\mathbf{p}}^{k-1})$.

- **(End of segment is best response)** $\sum_{k'=1}^k \tau^{k'} u_L\left(\mathbf{p}^{k'}, a^k\right) \geq \sum_{k'=1}^k \tau^{k'} u_L\left(\mathbf{p}^{k'}, a'\right)$ for any $a' \in [n]$. This represents the constraint $a^k \in \mathsf{BR}(\overline{\mathbf{p}}^k)$.

The objective of the LP is the optimizer utility $\sum_{k=1}^K \tau^k u_O(\mathbf{p}^k, a^k)$. If we want to further impose that the contract is a free-fall contract, we can add the constraint that $\mathbf{p}^k = 0$ for $k > 1$.

For free-fall contracts, we have an additional constraint on what sequences of actions are possible. Note that a free-fall contract will never repeat an action – in particular, after the initial segment, the cumulative utility of each action $i \in [n]$ decreases by $c_i$ per round, so the sequence of actions $(a^1, a^2, \ldots, a^K)$ a free-fall contract passes through must be sorted in *decreasing order of cost*. This means there are at most $2^n$ sequences of actions to check, and by checking all of them we can provably compute the optimal free-fall contract for a given general contract instance.

On the other hand, it's not clear if there are any constraints on how complex the sequence of actions for the optimal general dynamic contract can be – it is an interesting open question whether there exists any efficient (or even computable) algorithm for computing $U^\star$ for a general contract instance. Luckily, in order to show this separation, we need only exhibit a single general contract which outperforms the best free-fall contract. In the example above, we compute the best general contract for the same action sequence that the optimal free-fall contract passes through, and observe that the general contract obtains strictly larger utility.

---

[12]The code verifying this example can be found at: https://colab.sandbox.google.com/gist/jschnei/4d067ac2892d6b7c215dcea909577c53/optimal-dynamic-contracts-minimal-example.ipynb

# H  Simplifying Tool: Reductions from Discrete to Continuous Time

## H.1  Proof of Theorem 2.4

In this section we prove Theorem 2.4, showing that instead of working with discrete-time learning algorithms, it instead suffices to work with the set of continuous-time trajectories piecewise-linear trajectories described in Section 2.2. Our proof will generally follow the proof structure of [27] (which proves a similar reduction in the case of two-player bi-matrix games), with a few slight additional complexities due to some differences in notation (namely, we do not insist that every segment lies in the interior of a best-response region).

Before we begin the proof, it will be useful to establish a helpful auxiliary lemma about trajectories. Call a segment $(p^k, \tau^k, a^k)$ of a trajectory $\pi$ *degenerate* if it lies on the boundary of two best-response regions (i.e., $|\mathsf{BR}(\overline{\mathbf{p}}^{k-1}) \cap \mathsf{BR}(\overline{\mathbf{p}}^k)| \geq 2$), and *non-degenerate* otherwise. Let $\mathsf{Util}_0(\pi)$ be the utility contributed by just non-degenerate segments. We begin by showing that starting with any trajectory $\pi$, we can construct a mostly non-degenerate trajectory $\pi_0$ with $\mathsf{Util}_0(\pi_0)$ almost as large as $\mathsf{Util}(\pi)$.

**Lemma H.1.** *For any trajectory $\pi$ and any $\varepsilon > 0$, there exists a trajectory $\pi_0$ such that $\mathsf{Util}_0(\pi_0) \geq (1 - \varepsilon)\mathsf{Util}(\pi)$.*

*Proof.* Let $\pi = \{(p^k, \tau^k, a^k)\}$. We will produce $\pi_0$ by interleaving a sequence of small perturbations $(q^k, \delta^k)$ into $\pi$ for some $q^k \in \mathbb{R}_{\geq 0}^m$ and $\delta^k > 0$; that is, we will let $\pi_0$ be defined by the sequence of segments $(q^1, \delta^1), (p^1, \tau^1, a^1), (q^2, \delta^2), \ldots, (q^k, \delta^k), (p^k, \tau^k, a^k)$. Note that we have not specified the best-response of the learner for the perturbation segments $(q^k, \delta^k)$, because we will not count the utility from these segments (in fact, these perturbation segments might cross best-response boundaries, in which case we can split them into smaller segments). We will show that if we choose $q^i$ and $\delta^i$ correctly, $a^k$ is the unique best-response for each of the shifted $(p^k, \tau^k, a^k)$ segments.

Without loss of generality, assume $\sum_k \tau^k = 1$. For any $t \in [0, 1]$, we will let $\overline{\mathbf{p}}(t)$ be the average contract at time $t$ under trajectory $\pi$. That is, if $t = \tau^1 + \tau^2 + \cdots + \tau^{i-1} + \tau$ with $0 \leq \tau < \tau^i$, then

$$\overline{\mathbf{p}}(t) = \frac{\tau^1 p^1 + \tau^2 p^2 + \cdots + \tau^{i-1} p^{i-1} + \tau p^i}{t}.$$

For each $i \in [k]$, we will also let $\Delta^i = \delta^1 + \delta^2 + \cdots + \delta^i$, and $Q^i = (\delta^1 q^1 + \delta^2 q^2 + \cdots + \delta^i q^i)/\Delta^i$. Now, if $t = \tau^1 + \tau^2 + \cdots + \tau^{i-1} + \tau$ with $0 \leq \tau < \tau^i$, we will let $\overline{\mathbf{p}}_0(t)$ be the average contract under trajectory $\pi_0$ at time $\Delta^i + \tau$ (i.e., time $\tau$ into segment $(p^i, \tau^i, a^i)$). It is the case that for such $t$,

$$\overline{\mathbf{p}}_0(t) = \frac{Q^i \Delta^i + t\overline{\mathbf{p}}(t)}{\Delta^k + t} = \overline{\mathbf{p}}(t) + \frac{\Delta^i}{\Delta^i + t}(Q^i - \overline{\mathbf{p}}(t)).$$

We would like to choose $Q^i$ and $\Delta^i$ such that for each $i \in [k]$, for a large sub-interval of $\tau \in [0, \tau^i)$, the unique best response to $\overline{\mathbf{p}}_0(t)$ is exactly $a^i$. To begin, note that for any sequence of *strictly positive contracts* $Q^i \in \mathbb{R}_{>0}^m$, there is a sequence of $q^i$ and $\delta^i$ that implements it (because we can make each $Q^i$ any convex combination of $Q^{i-1}$ and $q^i$). Moreover, we can make $\Delta^k$ arbitrarily small, because scaling all the $\delta^i$ simultaneously does not affect the values of the $Q^i$.

Now, for each $i$, we will set $Q^i$ to a positive contract that uniquely incentivizes action $a^i$. Note that a non-negative contract exists by our assumption in Section 2; but since infinitesimal perturbations maintain the property that the contract uniquely incentivizes $a^i$, there must also be a positive contract with this property. We claim that if $\mathsf{BR}(Q^k) = \{a^k\}$ and $a^k \in \mathsf{BR}(\overline{\mathbf{p}}(t))$, then for any $\lambda \leq 1$, $\mathsf{BR}(\overline{\mathbf{p}}(t) + \lambda(Q^k - \overline{\mathbf{p}}(t))) = \{a^k\}$. To see this, note that we can write $\overline{\mathbf{p}}(t) + \lambda(Q^k - \overline{\mathbf{p}}(t)) = (1 - \lambda)\overline{\mathbf{p}}(t) + \lambda Q^k$. Since the utility of the agent is an affine linear function in the contract they are offered, for any action $a' \neq a$ we have that $u_A((1 - \lambda)\overline{\mathbf{p}}(t) + \lambda Q^k, a^k) = (1 - \lambda)u_A(\overline{\mathbf{p}}(t), a^k) + \lambda u_A(Q^k, a^k) > (1 - \lambda)u_A(\overline{\mathbf{p}}(t), a') + \lambda u_A(Q^k, a') = u_A((1 - \lambda)\overline{\mathbf{p}}(t) + \lambda Q^k, a^k)$.

It follows that if we choose the $Q^i$ in this way, $\mathsf{BR}(\overline{\mathbf{p}}_0(t)) = \{a^k\}$, and therefore each of the segments $(p^i, \tau^i, a^i)$ is non-degenerate. We will set $\Delta^k$ equal to $\varepsilon$. Doing so, we have that:

$$\mathsf{Util}_0(\pi_0) \geq \frac{\sum_{i=1}^{k} u_P(p^i, a^i)\tau^i}{1 + \Delta_k} \geq (1 - \varepsilon)\mathsf{Util}(\pi)$$

$\square$

Equipped with Lemma H.1, we can now prove Theorem 2.4.

*Proof of Theorem 2.4.* We follow the proof structure of [27] and prove both parts separately.

**Part 1.** Let $\pi = \{(p^k, \tau^k, a^k)\}_{k=1}^{K}$ represent a valid strategy for the principal in the continuous-time problem. Without loss of generality, assume $\sum_k \tau^k = 1$ (if not, we can divide all $\tau^k$ through by $\sum_k \tau_k$ without changing the value of this strategy). We will convert $\pi$ into the following discrete-time strategy for the principal: for each $i \in [K]$ (in order), the principal offers the contract $p^k$ for $\tau^k T$ rounds.

Our goal is to show that for any $\delta > 0$ and any mean-based algorithm $\mathcal{A}$, the above strategy results in at least $(\mathsf{Util}_0(\pi) - \delta)T - o(T)$ utility for the optimizer. The conclusion then follows by choosing a trajectory $\pi$ for which $\mathsf{Util}_0(\pi) > U^\star - \varepsilon/2$ (such a $\pi$ exists by Lemma H.1 and the definition of $U^\star$) and some $\delta < \varepsilon/2$. In the remainder of this proof, we will fix a specific mean-based algorithm $\mathcal{A}$ that is $\gamma(T)$-mean-based for some $\gamma(T) = o(1)$.

As in Definition 2.2, let $\sigma_{i,t}$ denote the aggregate utility of action $i$ to the agent over the first $t$ rounds. Let $T^k = \sum_{j=1}^{k} \tau^j T$, and consider the values of $\sigma_t$ for rounds $t \in [T^{k-1}, T^k]$ corresponding to the $k$th segment. Note that $\sigma_t$ is linear in this interval and so we can interpolate

$$\sigma_t = \frac{(t - T^{k-1})\sigma_{T^{k-1}} + (T^k - t)\sigma_{T^k}}{\tau^k T}. \tag{1}$$

Furthermore, assume that segment $k$ is non-degenerate, and so $\mathsf{BR}(\overline{\mathbf{p}}^{k-1}) \cap \mathsf{BR}(\overline{\mathbf{p}}^k) = \{a^k\}$. In particular, for any $t \in [T^{k-1}, T^k]$ and $a' \neq a_k$, either $\sigma_{T^{k-1}, a^k} > \sigma_{T^{k-1}, a'}$ or $\sigma_{T^k, a^k} > \sigma_{T^k, a'}$. As a consequence of (1), this means that for any $\varepsilon_k > 0$, there exists a $\delta_k > 0$ such that for $t \in [T^{k-1} + \varepsilon_k \tau^k, T^k - \varepsilon_k \tau^k]$, $\sigma_{t, a^k} \geq \sigma_{t, a'} + \delta_k T$. For sufficiently large $T$, $\delta_k T > \gamma(T)T$, and so the learner will put weight at least $(1 - n\gamma(T))$ on action $a^k$. The total utility of the principal from these rounds is therefore at least

$$(1 - n\gamma(T))(1 - 2\varepsilon_k)\tau^k u_P(p^k, a^k) \geq \tau^k u_P(p^k, a^k) - (n\gamma(T) + 2\varepsilon_k)T. \tag{2}$$

Summing over all non-degenerate segments $k$, we find the total utility of the principal is at least

$$\sum_k \tau^k u_P(p^k, a^k) - \sum_k k(n\gamma(T) + 2\varepsilon_k)T = \mathsf{Util}_0(\pi) - \sum_k k(n\gamma(T) + 2\varepsilon_k)T.$$

By choosing $\varepsilon_k$ sufficiently small, we can guarantee that this is at least $\mathsf{Util}_0(\pi) - \delta T$ for sufficiently large $T$, as desired.

**Part 2.** Fix any $\varepsilon > 0$. Assume that for some sufficiently large $T_0$, there exists a (possibly adaptive) dynamic strategy for the principal that guarantees utility at least $(U^\star + \varepsilon)T_0$ against every mean-based agent. We will show that this implies the existence of a continuous trajectory $\pi$ and $\mathsf{Util}(\pi) \geq U^\star + \varepsilon$, contradicting the definition of $U^\star$. Fix $\gamma(T) = T^{-1/2}$ and at any time $t$, let $J_t = \{j \in [n] | (\max_i \sigma_{t,i}) - \sigma_{t,j} < \gamma(T)T\}$ be the set of actions for the learner whose historical performance are within $\gamma(T)T$ of the optimally performing action. The set $J_t$ contains exactly the set of actions that the mean-based guarantee implies the agent must play with high probability. Our agent will do the following: if the principal is about to play contract $p^t$, the agent will play the action $j \in J_t$ that minimizes $u_L(p^t, j)$ (note that because we are tailoring the agent to this principal, we can do this).

Assume that this results in the principal playing the sequence of contracts $p^1, p^2, \ldots, p^{T_0}$. Consider the trajectory $\pi$ defined by the sequence of tuples $(p^1, 1/T_0), (p^2, 1/T_0), \ldots, (p^{T_0}, 1/T_0)$. In this

description of the trajectory, we've omitted the response action for the agent, which can be any best-response action for that segment. In fact, some segments may not be valid, as they start in one best response region and end in another; for those, we can subdivide them into however many parts are necessary to form a valid trajectory.

Now, note that the sub-segments corresponding to the step $(p^t, 1/T_0)$ only contain agent actions in the set $J_t$. This is since the agent utility at the start of this segment is $\sigma_t$, the agent utility at the end of this segment is $\sigma_{t+1}$, each component of $\sigma_{t+1} - \sigma_t$ is at most 1 (since the problem is bounded), but any action $j$ not in $J_t$ is at least $\gamma(T)$ away from optimal. The principal's utility contributed by this segment is therefore at least $\frac{1}{T_0} \min_{j \in J_t} u_P(p^t, j)$. But this is exactly the utility the principal obtained in round $t$ of the discrete-time game. Therefore the total utility $\mathsf{Util}(\pi)$ of this trajectory is at least $U^\star + \varepsilon$ – but this contradicts the definition of $U^\star$, as desired. $\qquad\square$

We will need the following lemma which says we can restrict our attention to finite-support $\mathcal{D}$ without loss of generality.

**Lemma H.2.** *Fix a principal-agent problem, a $\gamma > 1$, and an $\varepsilon > 0$. Let $\mathcal{D}$ be any distribution over trajectories. Then there exists a finite-support distribution $\mathcal{D}'$ over trajectories with the property that $\mathsf{Util}_\gamma(\mathcal{D}') \geq \mathsf{Util}_\gamma(\mathcal{D})$.*

*Proof.* We first claim the following: if a distribution $\mathcal{D}$ has the property that $\mathbb{E}_{\pi \sim \mathcal{D}}[u_P^{(t)}(\pi)] \geq Ut$ for each $t$ in the *discretized* set of time-intervals $S_{\varepsilon,\gamma} = \{1/\gamma, 1/\gamma + \varepsilon/\gamma, 1/\gamma + 2\varepsilon/\gamma, \dots, 1 - \varepsilon/\gamma, 1\}$, then it is the case that $\mathbb{E}_{\pi \sim \mathcal{D}}[u_P^{(t)}(\pi)] \geq (U - \varepsilon)t$ for all $t \in [1/\gamma, 1]$. This follows from the fact that the principal's profit per round is bounded above by 1, so $|u_P^{(t')}(\pi) - u_P^{(t)}(\pi)| \leq |t' - t|$. In particular, if $t'$ is the closest element of $S_{\gamma,\varepsilon}$ to a $t \in [1/\gamma, 1]$, it is the case that $|u_P^{(t')}(\pi) - u_P^{(t)}(\pi)| \leq \varepsilon/\gamma \leq \varepsilon t$.

Now, associate to each trajectory $\pi$ the $|S_{\varepsilon,\gamma}|$-tuple of real numbers $f(\pi) = \{u_P^{(t)}(\pi)\}_{t \in S_{\varepsilon,\gamma}}$; define $f(\mathcal{D}) = \mathbb{E}_{\pi \sim \mathcal{D}}[f(\pi)]$. Define $\mathcal{X} = \{f(\pi) \mid \pi \text{ is a trajectory}\} \subset \mathbb{R}^{|S|}$ to be the set of all such tuples. By Caratheodory's theorem, we can construct a distribution over at most $|S| + 1$ elements of $\mathcal{X}$ that (is arbitrarily close to) $f(\mathcal{D})$, for any $\mathcal{D}$. If we let $\mathcal{D}'$ be the corresponding distribution over trajectories, this satisfies the constraints of the theorem statement. $\qquad\square$

## H.2 Reduction from Discrete to Continuous Time with Unknown Time Horizons

In this section, we extend the previous reduction to the case where the time horizon can belong to an interval. One of the biggest differences is the introduction of this parameter $\gamma \geq 1$ which equals the multiplicative ratio $(\overline{T}/\underline{T})$. Instead of a trajectory $\pi = \{(p^k, \tau^k, a^k)\}_{k=1}^K$ being solely evaluated at its end time $\mathcal{T}^K$, we now care about its performance over its final interval of multiplicative width $\gamma$, namely $[\frac{1}{\gamma}\mathcal{T}^K, \mathcal{T}^K]$.

In order to quantify the performance of a trajectory at a certain time $t$, we will introduce some corresponding parenthetical superscript notation. In particular, $u_P^{(t)}(\pi)$ will denote the cumulative expected principal utility of trajectory $\pi$ from time zero to $t$, and is formally defined as

$$u_P^{(t)}(\pi) \triangleq \begin{cases} \sum_{k=1}^{k'-1} \tau^k u_P(p^k, a^k) + (t - \mathcal{T}^{k'}) u_P(p^{k'}, a^{k'}) & \text{if } t \in [\mathcal{T}^{k'}, \mathcal{T}^{k'+1}) \\ \sum_{k=1}^{k'-1} \tau^k u_P(p^k, a^k) & \text{if } t = \mathcal{T}^K. \end{cases}$$

Then, the worst-case (under possible time horizons) expected (under drawing from the distribution and actions producing random outcomes) utility of the principal for distribution $\mathcal{D}$ is given by

$$\mathsf{Util}_\gamma(\mathcal{D}) = \min_{x \in [1/\gamma, 1]} \mathbb{E}_{\pi \sim \mathcal{D}} \frac{u_P^{(x\mathcal{T}^K)}(\pi)}{x\mathcal{T}^K},$$

where each $\mathcal{T}^K$ is according to the drawn trajectory $\pi$.

Finally, let $U_\gamma^\star = \sup_\mathcal{D} \mathsf{Util}_\gamma(\mathcal{D})$, where the sup runs over all distributions of valid trajectories of arbitrary finite length. We can think of $U_\gamma^\star$ as the maximum possible worst-case utility of the principal in the unknown time horizon continuous setting game.

**Theorem H.3.** *Fix any principal-agent problem and $\gamma \geq 1$. We have the following two results:*

1. *For any $\varepsilon > 0$, there exists an oblivious strategy for the principal that gets at least $(U_\gamma^\star - \varepsilon)t - o(t)$ utility for the principal for all $t \in [T, \lceil \gamma T \rceil]$ for sufficiently large $T$.*

2. *For any $\varepsilon > 0$, there exists a mean-based algorithm $\mathcal{A}$ such that no (even adaptive[13]) principal can get more than $(U_\gamma^\star + \varepsilon)t + o(t)$ utility against an agent running $\mathcal{A}$ for all $t \in [T, \lceil \gamma T \rceil]$ for any $T$.*

*Proof of Theorem H.3.* **Part 1.** Begin by picking a strategy $\mathcal{D}$ for the optimizer in the continuous-time game that achieves utility at least $U_\gamma^\star - \varepsilon/2$. This strategy $\mathcal{D}$ is a distribution over trajectories $\pi$; by Lemma H.2, we can assume (at the cost of losing an arbitrarily small $O(\varepsilon)$ term) that $\mathcal{D}$ has finite support. For each of these trajectories, we apply Lemma H.1 to transform $\pi$ into a new trajectory $\pi'$ which obtains at least $(1 - \varepsilon)$ fraction of the utility of $\pi$ on non-degenerate segments. We will also normalize the total duration of each $\pi'$ to 1.

Now, note that since inequality (2) holds per segment (and indeed, even fractionally per segment), we can convert each resulting trajectory $\pi'$ to a discrete-time strategy over $\overline{T}$ rounds, with the property that for sufficiently large values of $\overline{T}$, for any $t \in [\underline{T} = \overline{T}/\gamma, \overline{T}]$, the utility of this discrete-time strategy until time $t$ is at least $u_P^{(t)}(\pi)$. Taking the corresponding distribution over these discrete-time strategies (choosing a sufficiently large $\overline{T}$ for all such strategies – note that we can do this because $\mathcal{D}$ has finite support), we obtain a discrete-time randomized (but otherwise oblivious) strategy for the principal that satisfies the theorem statement.

**Part 2.** As in the previous proof, fix an $\varepsilon > 0$, assume to the contrary there exists a $T_0$ along with a discrete-time (possibly randomized / adaptive) dynamic strategy which achieves at least $(U_\gamma^\star + \varepsilon)t$ utility for the principal for all $t \in [T_0/\gamma, T_0]$ against any mean-based bidder. Construct the same mean-based bidder as in the proof of part 2 of Theorem 2.4, which always picks the action in the set of approximate best-responses that least to the minimum expected utility for the principal.

When this principal plays against this agent, this leads to a distribution over sequences of contracts $(p^1, p^2, \ldots, p^{T_0})$. Each such sequence can be converted to a trajectory $\pi$ of the form $\{(p^1, 1/T_0), (p^2, 1/T_0), \ldots, (p^{T_0}, 1/T_0)\}$. This trajectory $\pi$ not only has the property that $\mathsf{Util}(\pi)T$ upper bounds the utility of the discrete-time agent (as in the proof of part 2 of Theorem 2.4), but in fact $u_P^{(t)}(\pi)$ is at least the utility of the agent discrete-time agent at time $tT_0$ (by exactly the same logic). It follows that if we let $\mathcal{D}$ be the distribution over such trajectories, it is the case that $\mathsf{Util}_\gamma(\mathcal{D}) \geq U_\gamma^\star + \varepsilon$. This contradicts the definition of $U_\gamma^\star$, as desired. □

Finally, we conclude this supplementary section with the proof of a preliminary lemma exploited in Section 3.1

**Lemma H.4.** *(Restated Lemma B.2) Consider any dynamic contract. For any time interval in which a mean-based agent plays a single action, we can replace the contracts in this interval with their average and obtain overall a revenue-equivalent dynamic contract.*

*Proof.* The result follows since the utility for the principal $u_P$ is affine in its first argument. Formally, let $\pi = \{(p^k, \tau^k, a^k)\}_{k=1}^K$ be a dynamic contract, with $a^k = a^{k+1} = a$ for some $k$. Consider a different contract $\pi'$ where we replace the consecutive pair of segments $(p^k, \tau^k, a^k)$ and $(p^{k+1}, \tau^{k+1}, a^{k+1})$ with the their average segment, i.e., $(\overline{p}, \tau^k + \tau^{k+1}, a)$, where $\overline{p} = (p^k \tau^k + p^{k+1} \tau^{k+1})/(\tau^k + \tau^{k+1})$, and all other segments remain the same as in $\pi$. Then, we have $\mathsf{Util}(\pi') - \mathsf{Util}(\pi) = \frac{1}{\sum_{k=1}^K \tau^k}\left(\tau^k u_P(p^k, a) + \tau^{k+1} u_P(p^{k+1}, a) - (\tau^k + \tau^{k+1})\overline{p}\right) = 0$. That is, both contracts give same utility for the principal. A similar argument holds for the discrete formulation of the model as well. □

---

[13]As with the known time-horizon result, this holds against adaptive principals in the full-feedback setting, or if the principal is deterministic. See Appendix H.3 for details.

### H.3 Mean-Based Algorithms in the Partial-Feedback Setting

We conclude with some clarifying remarks on the definition of a mean-based learning algorithm in a stochastic, partial-feedback setting (the bandits setting). The proofs of Theorems 2.4 and H.3 continue to hold essentially as written, but there are some subtleties that are worth pointing out.

We begin by clarifying the definition of mean-based in a partial-information setting. Formally, we write it as follows. Recall that $\sigma_i^t = \sum_{t'=1}^{t-1} u_i^{t'}$ is equal to the expected utility the learner would receive if they had played action $i$ for the first $t - 1$ rounds, *assuming the sequence of contracts the principal offers the learner remains static* (so in particular, for an adaptive / stochastic principal, $\sigma^t$ is a random variable).

**Definition H.5.** *(Mean-based algorithms in partial-information settings) A learning algorithm in a partial-information setting is $\gamma(T)$-mean-based if the following conditions hold: Fix any adaptive dynamic strategy of the principal and let (for each round $t \in [T]$), $X_t$ be the event that $\sigma_i^t < \sigma_{i'}^t - \gamma(T) \cdot T$, and $Y_t$ be the event that the algorithm takes action $i$ in round $t$. Then the algorithm is mean based if the probability $\Pr[X_t \wedge Y_t]$ (over all randomness in the learner's algorithm, principal's strategy, and problem setting) is at most $\gamma(T)$. We say an algorithm is mean-based if it is $\gamma(T)$-mean-based for some $\gamma(T) = o(1)$.*

The above definition is very similar to Definition 2.2; the main reason for stating it like this is to avoid implying the slightly stronger constraint that event $X_t$ deterministically implies that the probability of $Y_t$ is small conditioned on the current history of play. This implication is fine in the full-information setting where algorithms like multiplicative weights will indeed deterministically place small weight on action $i'$ if the event $X_t$ holds; but in the partial-information setting, there is always a chance that the learner is unable to accurately observe whether $X_t$ holds, and therefore no partial-information algorithm can achieve that guarantee. On the other hand, standard bandit algorithms with high-probability guarantees such as EXP3 (see [17]) satisfy the above definition of mean-based learning.

The proof of Theorem 2.4 works equally well with Definition H.5. The only subtlety is in Part 2, where to show a principal cannot do well against all mean-based agents, we design a mean-based agent that foils this specific principal. If the principal is randomized and adaptive, the agent cannot accurately predict the expected contract $p^t$ the principal will play in round $t$ (note that if the principal is adaptive but deterministic, the agent can still simulate the principal's behavior – likewise, if the principal is oblivious and randomized, the agent can compute the expected contract $p^t$ at any round). The proof of Theorem H.3 is similar.

## I Final Remarks

The following are observations about our repeated contract games with learning agents that arise from our analysis and from known results on learning agents in general games.

**Observation I.1.** *In the fixed contract setting, for any regret-minimizing agent in the limit $T \to \infty$ the support of the average empirical distribution of play includes only best-response actions with probability one. Therefore, the repeated game with a static contract against a regret-minimizing agent is essentially equivalent to the single-shot game against a rational agent.*

*Proof.* This follows directly from the regret-minimization property. Indeed, suppose, for the sake of contradiction, that there exists an action $a$ in the support which is not a best response. Denote the best-response utility by $OPT$. Action $a$ is played with probability $p > 0$. Notice that since there is only one player, the regret from any other action cannot be negative. Then we have that the regret is $\text{Regret} \geq p(OPT - u(a))T = \mathcal{O}(T)$, a contradiction. $\square$

**Observation I.2.** *If the agent is using a no-swap-regret algorithm, then the optimal static contract played repeatedly is also optimal in the dynamic setting. As a corollary, this is the case also for general no-regret algorithms if the agent has at most two actions.*

*Proof.* The result follows from [27], who show that in any game between an optimizer and a no-swap-regret algorithm, the optimizer cannot extract higher payoff than the Stackelberg value of the game where the optimizer plays the first move. The corollary is since with (at most) two actions internal regret and external regret are equivalent. $\square$

Below we show that in our analysis of dynamic linear contracts, it suffices to only examine linear contracts with $\alpha \in [0, 1]$. Note that although this is obvious in the static setting (offering $\alpha > 1$ requires the principal to suffer negative utility), it is not a priori clear that the principal cannot benefit via a dynamic strategy which offers a contract with $\alpha > 1$ for some fraction of the time horizon (perhaps counterbalancing it by offering a contract with a much smaller $\alpha$ later on). In fact, [43] show that when the agents have private information ("types") the principal *can* benefit by offering a randomized menu of linear contracts which possibly contains linear contracts with $\alpha > 1$.

Nonetheless, we show that the principal cannot benefit by doing this in the dynamic setting. The proof below follows from a slight modification of Lemma B.3 in our proof of Theorem 3.1.

**Observation I.3.** *Let $\pi = \{(\alpha^k, \tau^k, a^k)_{k=1}^K\}$ be any linear dynamic contract with some linear contract $\alpha^i > 1$. Then there exists a dynamic linear contract $\pi' = \{(\alpha^k, \tau^k, a^k)\}_{k=1}^K$ with $\mathsf{Util}(\pi') \geq \mathsf{Util}(\pi)$ and where $\alpha^k \leq 1$ for all k.*

*Proof.* We first observe that in Lemma B.3, when an agent is indifferent between actions $i$ and $i + 1$ then the change in utility for the principal by choosing an action $i + 1$ over $i$ is proportional to $(1 - \alpha^i)$. This is negative if $\alpha^i > 1$ and therefore the principal will prefer to agent to play action $i$ when $\alpha^i > 1$. However if $\alpha^i < 1$, then the principal will prefer that the agent play action $i + 1$. Thus in this modified rewriting lemma, contracts with breakpoints greater than 1, will prefer the lower action and breakpoints lower than 1, will prefer the higher action. By modifying Lemma B.3, we can rewrite any linear contract $\pi$ using the rewriting rules of Theorem 3.1 into a new linear contract $\pi'$ with a breakpoints that are at most 1, without any loss in utility. $\square$

