# OpenReview forum: "Contracting with a Learning Agent"
_NeurIPS.cc/2024/Conference — NeurIPS 2024 poster_

### Official Review · Reviewer_8ua3 · 2024-07-05

**Soundness:** 3
**Presentation:** 3
**Contribution:** 3
**Rating:** 6
**Confidence:** 3

**Summary:**

This theoretical paper studies repeated principal-agent contracts where the agent uses no-regret learning algorithms rather than complex strategic reasoning. The main results characterize optimal dynamic contracts against mean-based learning agents:

For linear contracts (including success/failure settings), the optimal dynamic contract has a simple "free-fall" structure - offer a carefully designed contract for some fraction of time, then switch to paying nothing. This can be computed efficiently.
There exist settings where both principal and agent benefit from the optimal dynamic contract compared to the best static contract.
With uncertainty about the time horizon, the principal's ability to outperform static contracts degrades as the uncertainty increases.

**Strengths:**

1. Novel and interesting problem formulation, bridging contract theory and online learning
2. Clean theoretical results with full proofs provided
3. Careful analysis of both linear and general contract settings
4. Considers practical issues like unknown time horizons
5. Results provide interesting insights, e.g. potential for "win-win" dynamic contracts

**Weaknesses:**

Limited to mean-based learning agents;
Optimal contracts for fully general (non-linear) settings not analyzed

**Questions:**

1. Are there natural economic settings where the "win-win" dynamic contracts might arise in practice?
2. Do you expect qualitatively similar results for multiple interacting agents? What are the key challenges there?
3. How do you expect the results to change if the agent uses a more sophisticated no-regret algorithm, such as one with bounded memory or one that is aware of the principal's strategy?
4. The paper focuses on optimizing the principal's utility. How would the analysis change if we consider Pareto-optimal contracts that balance utilities between the principal and agent?
5. Are there any interesting implications of your results for the design of real-world incentive structures, such as employee compensation plans or insurance contracts?
6. Your results show that dynamic contracts can sometimes benefit both parties. Are there conditions under which this is guaranteed, or conversely, conditions under which it's impossible?
7. The paper mentions potential extensions to MDPs. How do you envision applying these ideas to more complex sequential decision-making settings?

**Limitations:**

Yes

---

> ### Author Rebuttal · Authors · 2024-08-07
>
> Thank you for your feedback! We address the points raised in the review below.
>
> **Win-win dynamics:** In general, win-win situations arise in “general-sum” games, where the players are not complete adversaries, but rather can increase and share the overall welfare. In the constructions we used for demonstrating win-win dynamics, the structure was that different agent actions resulted in different levels of welfare. Thus, both players were better off if the principal could initially invest to create a strong motivation to play high-welfare actions. More broadly, one takeaway is that win-win scenarios, at least from our example, can be those where such an investment has an increasing return in terms of welfare.
>
> **Multiple agents:** The question of how to optimally incentivize multiple learning agents is an interesting one to consider in future work. A main challenge is that if agents jointly produce an output—meaning the payoff for the principal is a function of the agents’ joint action profile—it may become complicated to disentangle their contributions and incentivize them. It seems plausible that if agents’ contributions to the payoff are additive and contracts are linear (one alpha per agent), some multi-agent generalization of free-fall contracts might still be optimal. However, we have not analyzed this scenario. Other issues that may arise in interaction between multiple agents that one would need to consider in the model are multiplicity of equilibria and free-riding problems between the agents.
>
> **Different learning algorithms:** We agree that studying different types of learning approaches is interesting. Which type of learning is a good question. Some approaches may eliminate the possibility of free-fall contracts, but could have other limitations. We address below the two potential directions mentioned in the review. We also note that no-swap-regret is known to be the right notion for non-manipulability and results in the optimal static contract remaining optimal.
>
> *Bounded memory:* If we think about agents whose response at time $t+1$ is only a function of the history of steps $[t-k, t]$, with some constant $k$, other issues may arise, which are not necessarily in the benefit of the agent. This is related to the fact that such agents are not regret-minimizing. An extreme example is when $k=1$. Here, the principal can alternate between incentivizing some good action and paying zero in the following step. The principal ends up not paying anything , but the agent is always responding with a lag of one step and so he plays the good action half of the time. This kind of example is, in fact, not pathological, but could be extended to larger memory size.
>
> *Agent who is aware of the principals strategy:* In this case we need to think through how the agent reasons about their strategies in the repeated game, and are we still thinking about some notion of learning, or about general strategies in the repeated game. The latter case has been studied in prior literature, as we discuss in the introduction, and this was not our focus. A potentially related notion from online learning that one could think about in the context of the first case is contracting with an agent who is minimizing policy regret. This is an interesting direction to look into.
>
> **Pareto-optimal contracts:** The current solution, which maximizes the principal’s utility, is already Pareto-optimal (PO). If we consider PO contracts more generally, the analysis would include contracts that balance utilities between the principal and agent. We suspect that for the classic contracts setting, the entire PO curve can be recovered using similar linear programs to those used to compute the principal-utility-maximizing contract. When moving to the mean-based agent setting, our results already show that the PO curve is improved (we find a point with higher principal utility). However, there will be part of the PO curve that favors the agent and is bound by the total welfare, which cannot be improved. This is an interesting question to more carefully consider.
>
> **Real-world interpretation:** Our results could be interpreted in two main directions: from the perspective of the agent or the principal. On the agent’s side, a potential message is that in repeated contract scenarios, one should be careful in choosing which learning algorithms to implement. Using simple off-the-shelf options like Multiplicative Weights or Follow the Perturbed Leader may make the agent susceptible to exploitation by a sophisticated principal. On the principal’s side, the results show that one can do better by considering dynamic contracts, especially if there is reason to believe that the agent is using simple learning strategies or has some delay in response.
>
> **Conditions related to win-win scenarios:** We currently do not have a full characterization of the conditions for win-win scenarios. We can think of some such conditions. For example, “all actions have the same welfare” makes it impossible, because principal utility and agent utility sum to welfare and without increasing the welfare pie, principal gains come at the cost of agent losses. On the other hand, our analysis shows that the agent’s utility from a freefall contract is as if the final (stopping) action was played the entire time, so we know sufficient conditions need to imply that this stopping action has higher agent utility than the static contract’s induced action. This is an interesting question for future research.
>
> **Learning in contracts with MDP models:** While MDPs have been studied in contract settings. In the work we mention in the paper, the MDPs are either due to the principal having long-term constraints [reference 9 in the paper], or due to an underlying evolving state of the world [reference 53 in the paper], but not with learning agents. We have not considered the direction of learning agents with states. It would be an interesting direction for future research.

---

> > ### Comment · Reviewer_8ua3 · 2024-08-12
> > **Thanks for the response**
> >
> > Thanks for the response. I would keep my score for accepting this paper.

---

### Official Review · Reviewer_caMQ · 2024-07-11

**Soundness:** 4
**Presentation:** 4
**Contribution:** 3
**Rating:** 7
**Confidence:** 3

**Summary:**

The paper studies the repeated interaction between a principal and a learning agent. In particular, the authors assume that the agent employes a mean-based learning algorithm. The goal is to design a sequence of contracts that maximizes the principal’s cumulative utility. The main result of the paper is to show that in binary outcome settings the optimal strategy for the principal is to employ a “free-fall” contract in which: the principal commits to the same contract for some rounds and than switch to the zero contract. This result generalizes to settings with more than two outcomes in which the principal restricts to use linear contracts.

The second main result of the paper regards the uncertainty over the time horizon $T$. The authors show that this uncertainty hurts the effectiveness of dynamic contracts, characterizing the performance of optimal dynamic contracts.

**Strengths:**

The paper introduces a new interesting problem and provides interesting results. The techniques are novel and non-trivial. The paper is well-written.

**Weaknesses:**

I’ve some doubts on the realism of a model in which the agent is a no-regret minimizer, and not a swap regret minimizer. Nonetheless, this difference is clearly analyzed in the paper, and I do believe that the study of this setting is important.

**Questions:**

None.

**Limitations:**

None.

---

> ### Author Rebuttal · Authors · 2024-08-07
>
> Thank you for your feedback! We will further extend our discussion of the learning models, and in particular, the comparison between mean-based regret minimization and no-swap regret. Please see also our responses to the other reviews regarding this point.

---

> > ### Comment · Reviewer_caMQ · 2024-08-12
> >
> > Thanks for your response. I will keep my positive score.

---

### Official Review · Reviewer_x3aE · 2024-07-13

**Soundness:** 3
**Presentation:** 4
**Contribution:** 3
**Rating:** 6
**Confidence:** 4

**Summary:**

This paper considers the problem of contract design against a (mean-based) no-regret agent. The papers shows several results on the optimal contract design in this dynamic setting. First, with binary outcome, dynamic linear contract, it is optimal to design a free-fall contract. Second, the paper constructs instances (with non-zero measure) where optimal dynamic contract can pareto-improve the principal and agent utility than the optimal static contract. The paper also extends some of the results to the case of non-binary outcomes and unknown time horizon.

**Strengths:**

The paper is super well-written and easy to follow. It clearly explains the problem and the solution, as well as their relation to various lines of prior work. The examples and figures in these papers are also very carefully constructed to explain intuitions of their results. These readability optimizations help us a lot to get a quick and deep understanding of the paper.

**Weaknesses:**

While I think the paper is well-executed from its writing to technical derivation, the problem setting and the result it derives seems to me a bit artificial. The optimality of "free-fall contract" does not make any economic sense to me, but rather an unrealistic exploitation of the agent's no-regret learning algorithm. I can be wrong here, but I cannot think of any realistic situation in practice where anything similar to the free-fall contract is implemented. Maybe something loosely related: the quality of many restaurants, hotels over time can degrade after they establish a good reputation. This is because they can exploit the reputation to attract customers and reduce the quality to cut the cost. This is a kind of "free-fall contract" in a sense, but it is not sustainable (unless time horizon T is known to be finite like your setup). In general, I would say the mean-based regret model may not be a good model to capture the real-world learning agents, which prevents the unrealistic yet optimal solution of "free-fall contract": Perhaps they are not no-regret agents but admits a strong discounting factor to the historical reward in the regret notion so that they are quickly aware of the change in the contract (due to the distribution shift in the received payment) and adapt their response to it. The last section on the case of unknown time horizons is more realistic and it indeed rules out the free-fall contract as the optimal solution, though the results are not as strong as the simpler cases. Overall, I would say the paper could have taken a much stronger stance by exploiting its economic implications.

**Questions:**

Please address my concern in the above section if possible.

---

> ### Author Rebuttal · Authors · 2024-08-07
>
> Thank you for your feedback! We address the main point raised in the review below.
>
> **Mean-based learning and free-fall contracts:** We completely agree that—knowing now the exploitability of mean-based learning agents—studying different types of learning approaches, such as learning with recency bias or loss aversion, would be an important direction to explore. Such approaches can be realistic for human learners while preventing the principal from implementing free-fall contracts.
>
> There is, however, significant interest in establishing clear results for mean-based learning in our setting. First, we note that the result that a principal can exploit mean-based regret-minimizing learners in a large class of contract games using a very simple family of contracts (free fall) is not trivial and certainly not obvious before doing an analysis. Furthermore (see also our response to review HskF), our results show that it is not always better to use smarter learning strategies (after accounting for how the principal might adjust their dynamic contract). While this analysis (Theorem 3.2) uses a particular construction, it demonstrates that whether it is beneficial to use learning and which type of learning to use is more subtle and depends on the details of the game.
>
> Additionally, as also suggested in the review, with human agents, one could think of free-fall contracts as happening over at least limited time spans. There are lines of work in economics studying delayed or insufficient responses to new information (see [1,2,3] below), or even limited learning over extended time periods [4]. Essentially, a decision that was extremely good for a long period of time may have momentum and will not always be abandoned in the first (say) month of a bad outcome. Mean-based regret minimization is a form of learning with these properties.
>
> We would like to emphasize, however, that we think there is value in our analysis regardless of whether mean-based regret minimization is the way human agents act. Mean-based regret minimization is a prominent algorithmic approach to learning in repeated games, which has been extensively studied in other contexts. In particular, algorithms of this family have been traditionally motivated in the theoretical economics literature as “simple,” “adaptive,” and “natural” theoretical models of learning by boundedly rational agents (see, e.g., [5] and references therein). It is thus important to understand the implications of such learning approaches for repeated contracts, especially in comparison to other types of learning and to static contract setting. In this sense, our analysis of mean-based and no-swap regret learners in this paper is a first step in exploring contracts with learning agents more broadly.
>
> [1] Carroll, C.D., Crawley, E., Slacalek, J., Tokuoka, K. and White, M.N., 2020. Sticky expectations and consumption dynamics. American economic journal: macroeconomics, 12(3), pp.40-76.
>
> [2] Bouchaud, J.P., Krueger, P., Landier, A. and Thesmar, D., 2019. Sticky expectations and the profitability anomaly. The Journal of Finance, 74(2), pp.639-674.
>
> [3] Ba, C., Bohren, J.A. and Imas, A., 2022. Over-and underreaction to information. Available at SSRN 4274617.
>
> [4] Benjamin, D.J., Rabin, M. and Raymond, C., 2012. A Model of Non-belief in the Law of Large Numbers. Available at SSRN 1945916.
>
> [5] Hart, S. and Mas-Colell, A., 2013. Simple adaptive strategies: from regret-matching to uncoupled dynamics (Vol. 4). World Scientific.

---

> > ### Comment · Reviewer_x3aE · 2024-08-13
> >
> > Thanks for your response! I will keep my score unchanged.

---

### Official Review · Reviewer_HskF · 2024-07-13

**Soundness:** 4
**Presentation:** 4
**Contribution:** 3
**Rating:** 7
**Confidence:** 4

**Summary:**

This work studies the problem of contracting with a no-regret learning agent. They show that

- In linear contracts, the optimal dynamic contract against a mean-based learning agent is a free-fall contract.
- dynamic contracts can be win-win. Both principal and agent can benefit from some dynamic contractor and mean-based learning agent.
- Knowing time horizon is very important. They provide a lower bound showing that there exists a problem in which no dynamic strategies outperform the optimal static linear contract.

**Strengths:**

The problem studied by this work is very interesting --- repeated contracts with learning agents.

The results are novel. There are some interesting findings in this work, including the optimality of free-fall contract, the existence of win-win scenarios, and the impact of the knowledge time horizon.

The writing is clear.

**Weaknesses:**

If I have to say some weaknesses of this work, I would say most results seem to be instance-dependent but not general.

**Questions:**

- In line 218-226, they introduced full feedback and bandit feedback. I am a bit confused here about bandit feedback. Does this mean that the agent won't observe the chosen contract $p_t$ at each round?

- In Thm 3.2, they show that there exists a problem in which optimal dynamic contracts lead to "win-win" for both principal and agent. I am curious if there are scenarios in which agents get hurt by running learning algorithms.

- For Thm 4.3, in the construction for proving the theorem, is $(\epsilon, 1)$ feasible? I think they should mention this to justify that the infeasibility is indeed caused by the uncertainty of time horizon but not that the game itself is infeasible.

---

> ### Author Rebuttal · Authors · 2024-08-07
>
> Thank you for your feedback! We address the points raised in the review below.
>
> **Bandit feedback:**  One possibility for bandit feedback is when the agent does not observe the contract $p_t$ but only observes the payoff induced by this contract for the action that was played. A different scenario is when the agent may observe the contract but does not know the outcome distributions or what they imply for the actions that were not played, which again requires some additional exploration. Our analysis, however, holds for full feedback or any of these partial feedback scenarios.
>
> **Theorem 3.2:** The theorem shows that there is a positive measure of contract games where learning by the agent and an optimal dynamic contract by the principal lead to a strict Pareto improvement (win-win) outcome. This is, however, not true for all contract games; there are games where learning reduces a player’s utility compared to a myopic best response (for instance, the example in the introduction). The construction in Theorem 3.2 shows that there are cases where using either a best-response strategy or, alternatively, smarter learning (no-swap-regret) leads to a loss compared to using simpler learning (mean-based learning; see our remark at the end of Section 3). So, whether it is beneficial to use learning, and which type of learning to use, depends on the game.
>
> **Theorem 4.3:** Whether or not $(\epsilon,1)$ is feasible just depends on whether $(1 + \epsilon)$ is less than the ratio between the optimal (known-time horizon) dynamic contract and the optimal static contract; for any problem there is some sufficiently large $\epsilon$ to make $(\epsilon, 1)$ infeasible. The theorem statement is perhaps easier to conceptualize by considering its contrapositive: if there is an $\epsilon > 0$ such that $(\epsilon, 1)$ is feasible, then for any $\gamma > 1$ there exists an $\epsilon_\gamma$ such that $(\epsilon_\gamma, \gamma)$ is feasible as well. We will provide additional context for this theorem to better clarify the situation.

---

> > ### Comment · Reviewer_HskF · 2024-08-13
> >
> > Thanks for your response! I will keep my score unchanged.

---

### Author Rebuttal · Authors · 2024-08-07

We thank the reviewers for the constructive comments and will use this feedback to improve our paper. We address the specific points raised in the reviews in a separate response to each review.

---

### Decision · Program_Chairs · 2024-09-25

**Decision:**

Accept (poster)

**Comment:**

The paper studies repeated contracts with learning agents. The problem studied is novel, and the derived results are comprehensive, characterizing optimal dynamic contracts against a mean-based learner. The reviewers agree on the clarity of writing, indicating that the paper is well-written and easy to follow. Given the novelty and the high quality of this work, I recommend acceptance.